# Validation of the accuracy of the modified World Federation of Neurosurgical Societies subarachnoid hemorrhage grading scale for predicting the outcomes of patients with aneurysmal subarachnoid hemorrhage

Tuan Anh Nguyen[1,2], Ton Duy Mai[2,3,4], Luu Dang Vu[5,6], Co Xuan Dao[2,4,7]*, Hung Manh Ngo[4,8,9], Hai Bui Hoang[2,10], Tuan Anh Tran[4,5,6], Trang Quynh Pham[9,11], Dung Thi Pham[12], My Ha Nguyen[13], Linh Quoc Nguyen[1,2], Phuong Viet Dao[2,3,4], Duong Ngoc Nguyen[1], Hien Thi Thu Vuong[14], Hung Dinh Vu[10], Dong Duc Nguyen[8], Thanh Dang Vu[15], Dung Tien Nguyen[3], Anh Le Ngoc Do[1,2], Cuong Duy Nguyen[16], Son Ngoc Do[2,4,7], Hao The Nguyen[9,11], Chi Van Nguyen[1,2], Anh Dat Nguyen[1,2], Chinh Quoc Luong[1,2,4]

1 Center for Emergency Medicine, Bach Mai Hospital, Hanoi, Vietnam, 2 Department of Emergency and Critical Care Medicine, Hanoi Medical University, Hanoi, Vietnam, 3 Stroke Center, Bach Mai Hospital, Hanoi, Vietnam, 4 Faculty of Medicine, University of Medicine and Pharmacy, Vietnam National University, Hanoi, Vietnam, 5 Radiology Centre, Bach Mai Hospital, Hanoi, Vietnam, 6 Department of Radiology, Hanoi Medical University, Hanoi, Vietnam, 7 Center for Critical Care Medicine, Bach Mai Hospital, Hanoi, Vietnam, 8 Department of Neurosurgery II, Neurosurgery Center, Vietnam-Germany Friendship Hospital, Hanoi, Vietnam, 9 Department of Surgery, Hanoi Medical University, Hanoi, Vietnam, 10 Emergency and Critical Care Department, Hanoi Medical University Hospital, Hanoi Medical University, Hanoi, Vietnam, 11 Department of Neurosurgery, Bach Mai Hospital, Hanoi, Vietnam, 12 Department of Nutrition and Food Safety, Faculty of Public Health, Thai Binh University of Medicine and Pharmacy, Thai Binh, Vietnam, 13 Department of Health Organization and Management, Faculty of Public Health, Thai Binh University of Medicine and Pharmacy, Thai Binh, Vietnam, 14 Emergency Department, Vietnam-Czechoslovakia Friendship Hospital, Hai Phong, Vietnam, 15 Emergency Department, Agriculture General Hospital, Hanoi, Vietnam, 16 Department of Emergency and Critical Care Medicine, Thai Binh University of Medicine and Pharmacy, Thai Binh, Vietnam

* daoxuanco@gmail.com

## Abstract

### Background

Evaluating the prognosis of patients with aneurysmal subarachnoid hemorrhage (aSAH) who may be at risk of poor outcomes using grading systems is one way to make a better decision on treatment for these patients. This study aimed to compare the accuracy of the modified World Federation of Neurosurgical Societies (WFNS), WFNS, and Hunt and Hess (H&H) Grading Scales in predicting the outcomes of patients with aSAH.

### Methods

From August 2019 to June 2021, we conducted a multicenter prospective cohort study on adult patients with aSAH in three central hospitals in Hanoi, Vietnam. The primary outcome was the 90-day poor outcome, measured by a score of 4 (moderately severe disability) to 6

**Data Availability Statement:** All relevant data are within the paper and its Supporting information files.

**Funding:** The author(s) received no specific funding for this work.

**Competing interests:** The authors have declared that no competing interests exist.

(death) on the modified Rankin Scale (mRS). We calculated the areas under the receiver operator characteristic (ROC) curve (AUROCs) to determine how well the grading scales could predict patient prognosis upon admission. We also used ROC curve analysis to find the best cut-off value for each scale. We compared AUROCs using Z-statistics and compared 90-day mean mRS scores among intergrades using the pairwise multiple-comparison test. Finally, we used logistic regression to identify factors associated with the 90-day poor outcome.

## Results

Of 415 patients, 32% had a 90-day poor outcome. The modified WFNS (AUROC: 0.839 [95% confidence interval, CI: 0.795–0.883]; cut-off value$\geq$2.50; $P_{AUROC}$<0.001), WFNS (AUROC: 0.837 [95% CI: 0.793–0.881]; cut-off value$\geq$3.5; $P_{AUROC}$<0.001), and H&H scales (AUROC: 0.836 [95% CI: 0.791–0.881]; cut-off value$\geq$3.5; $P_{AUROC}$<0.001) were all good at predicting patient prognosis on day 90th after ictus. However, there were no significant differences between the AUROCs of these scales. Only grades IV and V of the modified WFNS (3.75 [standard deviation, SD: 2.46] vs 5.24 [SD: 1.68], p = 0.026, respectively), WFNS (3.75 [SD: 2.46] vs 5.24 [SD: 1.68], p = 0.026, respectively), and H&H scales (2.96 [SD: 2.60] vs 4.97 [SD: 1.87], p<0.001, respectively) showed a significant difference in the 90-day mean mRS scores. In multivariable models, with the same set of confounding variables, the modified WFNS grade of III to V (adjusted odds ratio, AOR: 9.090; 95% CI: 3.494–23.648; P<0.001) was more strongly associated with the increased risk of the 90-day poor outcome compared to the WFNS grade of IV to V (AOR: 6.383; 95% CI: 2.661–15.310; P<0.001) or the H&H grade of IV to V (AOR: 6.146; 95% CI: 2.584–14.620; P<0.001).

## Conclusions

In this study, the modified WFNS, WFNS, and H&H scales all had good discriminatory abilities for the prognosis of patients with aSAH. Because of the better effect size in predicting poor outcomes, the modified WFNS scale seems preferable to the WFNS and H&H scales.

## Introduction

Subarachnoid hemorrhage (SAH) is often a devastating clinical event with a substantial mortality rate, a high rate of morbidity among survivors, and a healthcare burden [1, 2]. The most cause of spontaneous SAH is a ruptured aneurysm. The degree of neurologic impairment and the extent of subarachnoid bleeding at the time of admission are the most important predictors of neurologic complications and outcomes [3, 4]. Moreover, appropriate therapy for an aneurysmal SAH is often dependent upon the severity of the hemorrhage involved. Therefore, the significance and vitality of obtaining the SAH severity grade as soon as the presentation and stabilization of the patient allow that cannot be underestimated. In order to standardize the clinical classification of patients with SAH, several grading systems are utilized upon the initial evaluation–most notable of which are the grading systems proposed by Hunt and Hess (H&H) and that of the World Federation of Neurological Surgeons (WFNS) [5, 6].

To date, no grading systems are completely free from interobserver variability. For the H&H scale, the classifications are arbitrary, some of the terms are vague, and some patients may present with initial features that defy placement within a single grade [7]. As a result, the

interobserver agreement for the H&H scale is poor [8, 9]. A systematic review of SAH grading scales also found conflicting data regarding the utility of the H&H scale for prognosis [7]. Furthermore, it is unclear if there are significant differences in outcome for adjacent H&H grades [10–13]. Unlike the H&H scale, the WFNS scale uses objective terminology to assign grades [7]. Because the WFNS scale requires only an assessment of the Glasgow coma scale (GCS) and motor function, it may be easier to administer than the H&H scale. However, the lack of formal validation of the WFNS scale might lead to occasional overlap between grades (particularly between grades II and III), where the outcomes predicted by the assigned grade may not differ substantially [11, 14, 15]. As a result, the interobserver variability for the WFNS scale is still moderate [9]. Additionally, a systematic review of SAH grading scales also found conflicting data regarding the prognostic power of the WFNS grades [7]. Therefore, making more accurate initial predictions of outcome after SAH remains a challenge.

Stratification of patients is usually done in accordance to their clinical SAH grade prior to undergoing aneurysm repair during the acute phase [16–24]. Although it is very unlikely the SAH grading scale would yield a 100% accurate prediction of outcomes, in order to achieve the maximum intended effect, the ideal SAH grading can be obtained via the following prerequisites: 1) the scale must be easy to apply during the acute phase of the disease; 2) it must be free from observer variability; 3) there must be a significant correlation between patient outcome and the grading scale; 4) two adjacent grades must have significantly different outcomes [25]. Removing the presence of motor deficit, the WFNS had proposed a modification to the original WFNS scale in conjunction with the Japan Neurosurgical Society. In two studies, this modified WFNS scale is seen with a much more promising discriminatory value compared with the original WFNS scale, but broader validation studies are required [15, 26, 27].

The aim of this study was to determine the relationship between the grades on the modified WFNS, WFNS, and H&H scales and the actual outcome and to compare the prognostic accuracy of these scales.

## Methods

### Source of data

This multicenter prospective observational study is the major update of our published previous study [28–30], which collected data on patients with aneurysmal SAH consecutively admitted to the three national tertiary hospitals (Vietnam-Germany Friendship, Bach Mai, and Hanoi Medical University Hospital) in Hanoi, Vietnam, between August 2019 and August 2020, to investigate the rate of poor outcomes and associated factors from aneurysmal SAH in the country [28]. To determine the relationship between the grades on the modified WFNS, WFNS, and H&H scales and the actual outcome and to compare the prognostic accuracy of these scales, we continued to collect data on these patients consecutively admitted to these three hospitals between September 2020 and June 2021. We then merged the data sets from two stages of data collection for the three hospitals. These hospitals are designated central hospitals in northern Vietnam by the Ministry of Health of Vietnam; the first is a surgical hospital with 1,500 beds, the second is a large general hospital with 3,200 beds, and the last is a small hospital with 580 beds. Each participating hospital had at least two representatives (i.e., fully trained clinicians or surgeons) who were a part of the study team. Participation was voluntary and unfunded. All patients received a follow-up till death in the hospital or within 30 or 90 days of ictus and had clinic visits or phone contacts on days 30[th] and 90[th] after ictus for the modified Rankin Scale (mRS) assessments, mRS ranges from 0 (no disability) to 6 (death) [31], and evaluation of complications (e.g., chronic hydrocephalus).

## Participants

This study included all patients (aged 18 years or older) presenting with aneurysmal SAH to the three central hospitals within 4 days of ictus. We defined a case of aneurysmal SAH as a person who had the presence of blood visible on a head computed tomography (CT) scan (or in case the CT scan was negative, the presence of xanthochromia in the cerebral spinal fluid) in combination with an aneurysm confirmed on CT or digital subtraction angiography (DSA) [4]. We excluded patients for whom the GCS on admission was unable to be scored (e.g., patients intubated and under sedation before arrival at the central hospital) or patients who became lost at 90 days of follow-up during the study. In the case of aphasia, patients were classified according to the clinically possible GCS scores derived from their eye and motor scores [32, 33]. When different possible verbal scores placed patients in different categories, these patients were excluded.

All patients were managed following the American Heart Association (AHA)/American Stroke Association (ASA) guidelines for the management of aneurysmal SAH [4]. Aneurysm repair with endovascular coiling or surgical clipping was performed as early as possible and immediately if rebleeding occurred. The decision to treat the cerebral aneurysms was made based on the discretion of the physician in charge of the patients and the availability of endovascular coiling or neurosurgical clipping, which depended on the participating hospital and the financial situation (either insurance or patient self-pay).

## Data collection

The data for each study patient were recorded from the same unified samples (case record form). A case record form (CRF) was adopted across the study sites to collect the common variables. Data were entered by a researcher or investigator into the study database via EpiData Entry software (EpiData Association, Denmark, Europe), which was used for simple or programmed data entry and data documentation that could prevent data entry errors or mistakes. We also checked the data for implausible outliers and missing fields and contacted hospital representatives for clarification. Patient identifiers were not entered into the database to protect the patients' confidentiality.

## Outcome measures

The primary outcome of this study was poor neurological function (poor outcome) on day 90[th] after ictus, which was defined as mRS scores of 4 (moderately severe disability) to 6 (death) [34, 35]. We also examined the following secondary outcomes: poor outcome on day 30[th] after ictus, 30- and 90-day mortality rates, and incidence rate of complications.

## Predictor measures

We defined exposure variables as SAH grading scales (i.e., the PAASH, WFNS, and H&H grading scales) at the time of admission to the hospital. Based on the admission GCS, we divided patients into the five categories of the WFNS grading scale ranging from grade I (GCS score of 15) to V (GCS scores of 3 to 6), of which focal deficits make up 1 additional grade for patients with a GCS score of 14 or 13 [6], and into the five categories of the modified WFNS ranging from grade I (GCS score of 15) to V (GCS scores of 3 to 6), regardless of the presence of neurologic deficits [15]. Based on the clinical condition on admission, we also classified patients into the five severity groups according to the H&H grading scale, which consists of five grades ranging from minimally symptomatic to coma [5]. All data elements required for calculating the GCS score and for classifying patients according to the modified WFNS,

WFNS, or H&H grading scale at the time of admission to the hospital were prospectively assessed and collected on the same unified CRF by a fully trained clinician or surgeon of the participating hospitals and then were entered by a researcher or investigator into a study database via the EpiData Entry software for later analysis.

We determined confounding factors as variables collected on the same unified CRF by a fully trained clinician or surgeon. The CRF included variables based on the unruptured intracranial aneurysm (UIA) and SAH work group (WG) recommendations [36], such as information on:

(i) Medical histories (e.g., stroke, UIA, etc.), clinical presentation (e.g., GCS and focal neurological signs).

(ii) Admission head CT scan (e.g., presence of SAH, intraventricular hemorrhage (IVH) or intracerebral hemorrhage (ICH), and Fisher scale) and follow-up head CT scan during hospitalization (e.g., presence of SAH, IVH or ICH) or on days 30th and 90th after ictus (e.g., the presence of chronic hydrocephalus). We also collected data on the aneurysm site and aneurysm size from DSA or multi-slice CT (MSCT) angiography scan.

(iii) Surgical and endovascular interventions (i.e., surgical clipping or endovascular coiling), rescue therapies (e.g., surgical hematoma evacuation, defined as any surgical procedure evacuating epidural, subdural, intraventricular, or intraparenchymal hematoma, such as decompressive craniotomy, open craniotomy, or minimally invasive surgery; external ventricular drain (EVD) placement; ventriculoperitoneal (VP) shunt), and intensive care unit (ICU) therapies (e.g., mechanical ventilation).

(iv) Neurological complications (e.g., rebleeding, which included bleeding into the subarachnoid space, intracerebral, intraventricular, or subdural compartments; delayed cerebral ischaemia (DCI), hydrocephalus). Rebleeding from a ruptured aneurysm was classified into two subtypes: early or late rebleeding. We defined early or late rebleeding as rebleeding occurring in the hospital before or after an aneurysm repair, respectively.

(v) Clinical time course (e.g., time from ictus to hospital arrival, length of hospitalization).

(vi) We also collected data on demographics (i.e., sex, age) and system variables, which are available as an online supplement of a previously published paper [28].

## Sample size

In the present study, poor neurological function on day 90th after the ictus served as the primary outcome. We, therefore, used the formula to determine the minimal sample size for estimating a population proportion with a confidence level of 95%, a confidence interval (margin of error) of ±4.7% and an assumed population proportion of 39.1%, based on the rate of poor neurological function on day 90th after the ictus (39.1%) reported in a previously published study [37]. Therefore, we should have at least 415 patients in our sample. Because of this, our sample size was sufficient and reflected a normal distribution.

$$n = \frac{z^2 x \hat{p}(1 - \hat{p})}{\varepsilon^2}$$

where:
*z is the z score (z score for a 95% confidence level is 1.96)*
*ε is the margin of error (ε for a confidence interval of ± 4.7% is 0.047)*
*p̂ is the population proportion (p̂ for a population proportion of 39.1% is 0.391)*
n is the sample size

## Statistical analyses

We used IBM$^{\circledR}$ SPSS$^{\circledR}$ Statistics 22.0 (IBM Corp., Armonk, United States of America) and Analyse-it statistical software (Analyse-it Software, Ltd., Leeds, United Kingdom) for data analysis. We report the data as numbers (no.) and percentages (%) for categorical variables and medians and interquartile ranges (IQRs) or means and standard deviations (SDs) for continuous variables. Furthermore, comparisons were made between poor and good outcomes on day 30[th] and 90[th] after ictus for each variable using the Chi-squared test or Fisher's exact test for categorical variables and the Mann–Whitney U test, Kruskal–Wallis test, or one-way analysis of variance for continuous variables.

Odds ratios (ORs) for a poor outcome on days 30[th] and 90[th] after ictus with 95% confidence intervals (CIs) were calculated for each grade of the SAH grading scales (i.e., the modified WFNS, WFNS, and H&H scales) with a univariable logistic regression model, with grade I taken as the reference. In all of the SAH grading scales, significant intergrade differences concerning the outcome (mean mRS scores) on days 30[th] and 90[th] after ictus that were determined using the Kruskal–Wallis H test with the Dunn-Bonferroni principle as a post hoc analysis.

We converted from descriptive SAH grading scales to numerical SAH grading scales in ascending order (S1 Table in S1 File). Receiver operator characteristic (ROC) curves were plotted, and the areas under the ROC curve (AUROC) were calculated to determine the discriminatory ability of all SAH grading scales for the prognosis of the patients upon admission. The cut-off value of each SAH grading scale was determined by ROC curve analysis and defined as the cut-off point with the maximum value of Youden's index (i.e., sensitivity + specificity—1). Based on the cut-off value of each SAH grading scale, we assigned the patients to two severity groups: either the grade that was less than the cut-off value or another that was greater than or equal to the cut-off value. We also performed a pairwise comparison among the AUROCs of the SAH grading scales for predicting the poor outcome on days 30[th] and 90[th] after ictus by using the Z-statistics.

We assessed the factors associated with 90-day poor outcomes using logistic regression analysis. To reduce the number of predictors and the multicollinearity issue and resolve the overfitting, we used different methods to select variables as follows: (a) we put all variables (including exposure and confounding factors) of demographics, baseline characteristics, clinical and laboratory characteristics, neuroimaging findings, clinical time course, treatments, and complications into the univariable logistic regression model; (b) we selected variables if the p value was <0.05 in the univariable analysis between the good and poor outcomes on day 90[th] after ictus, as well as those that are clinically crucial, to put in the multivariable logistic regression model. These variables included demographics (i.e., age), risk factors for aneurysmal SAH (i.e., hypertension), comorbidities (i.e., diabetes mellitus), initial neuroimaging findings (i.e., location of blood within the subarachnoid space, the occurrence of IVH, and ICH, and aneurysm location), the severity of the aneurysmal SAH on admission (i.e., the grade of modified WFNS, WFNS, or H&H scale that was either greater than or equal to the cut-off value), treatments (i.e., aneurysm repair, nimodipine, surgical hematoma evacuation, EVD), and complications (i.e., rebleeding, DCI, acute hydrocephalus, and pneumonia). Using a stepwise backwards elimination method, we started with the full multivariable logistic regression model that included the selected variables. This method then deleted the variables stepwise from the full model until all remaining variables were independently associated with the risk of 90-day poor outcomes in the final model. Similarly, we used these methods of variable selection and analysis for assessing factors associated with 30-day poor outcomes. For examining the effect size of each grade of the SAH grading scales, in combination with confounding factors, for predicting the 30- and 90-day poor outcomes, we replaced the severity variable with each SAH

grading scale (i.e., the modified WFNS, WFNS, or H&H scale, with grade I taken as the reference) in the multivariable logistic regression models, with the same set of confounding variables. We presented the odds ratios (ORs) and 95% confidence intervals (CIs) in the univariable logistic regression model and the adjusted ORs (AORs) and 95% CIs in the multivariable logistic regression model.

For all analyses, the significance levels were two-tailed, and we considered P < 0.05 to be statistically significant.

### Ethical issues

The Hanoi Medical University (Approval number: 3335/QĐ-ĐHYHN), Vietnam-Germany Friendship Hospital (Approval number: 818/QĐ-VĐ; Research code: KH04.2020), and Bach Mai Hospital (Approval number: 3288/QĐ-BM; Research code: BM_2020_1247) Scientific and Ethics Committees approved this study. This study was conducted according to the principles of the Declaration of Helsinki. The Vietnam-Germany Friendship Hospital Scientific and Ethics Committees waived written informed consent for this non interventional study, and public notification of the study was made by public posting. The authors who performed the data analysis kept the datasets in password-protected systems, and we only present anonymized data.

## Results

Data on 415 eligible patients with aneurysmal SAH were submitted to the study database (Fig 1 and Table 1), in which there were few missing data.

### Baseline characteristics and clinical outcomes

Of the total patients, 198/415 (47.7%) were men, and the median age was 57.0 (IQR: 48.0–67.0) (Table 1). Overall, 32.0% (133/415) of patients with aneurysmal SAH had a poor outcome on day 90th after ictus, 23.4% (97/415) of whom died within 90 days of ictus (Table 1). The baseline characteristics, management, complications, and outcomes of the patients were compared between patients who had a good outcome and patients who had a poor outcome on days 30th and 90th after ictus, as shown in S2 to S9 Tables in S1 File.

### Overall prognostic performance of the SAH grading scales

Figs 2 and 3 show the overall performances of the SAH grading scales for predicting the poor outcome, of which the modified WFNS (AUROC: 0.839 [95% CI: 0.795–0.883]; cut-off value≥2.50; sensitivity: 82.7%; specificity: 77.7%; $P_{AUROC}$<0.001), the WFNS (AUROC: 0.837 [95% CI: 0.793–0.881]; cut-off value≥3.5; sensitivity: 75.9%; specificity: 83.0%; $P_{AUROC}$<0.001), and the H&H scales (AUROC: 0.836 [95% CI: 0.791–0.881]; cut-off value≥3.5; sensitivity: 72.2%; specificity: 84.4%; $P_{AUROC}$<0.001) all had good discriminatory abilities for predicting the 90-day poor outcome (Fig 3). There were also the good discriminatory abilities of the SAH grading scales for predicting the 30-day poor outcome on day 30th after ictus, as shown in Fig 2 and S10 Table in S1 File.

Table 2 shows the differences between the AUROC curves among different test-pairwise, of which the AUROCs for predicting the 90-day poor outcome did not differ significantly between the modified WFNS and WFNS scales (AUROC difference: 0.002; 95% CI: -0.001–0.005; Z-statistic: 1.25; p = 0.211), the modified WFNS and H&H scales (AUROC difference: 0.003; 95% CI: -0.015–0.021; Z-statistic: 0.27; p = 0.786), and the WFNS and H&H scales (AUROC difference: 0.001; 95% CI: -0.017–0.018; Z-statistic: 0.07; p = 0.947). For predicting

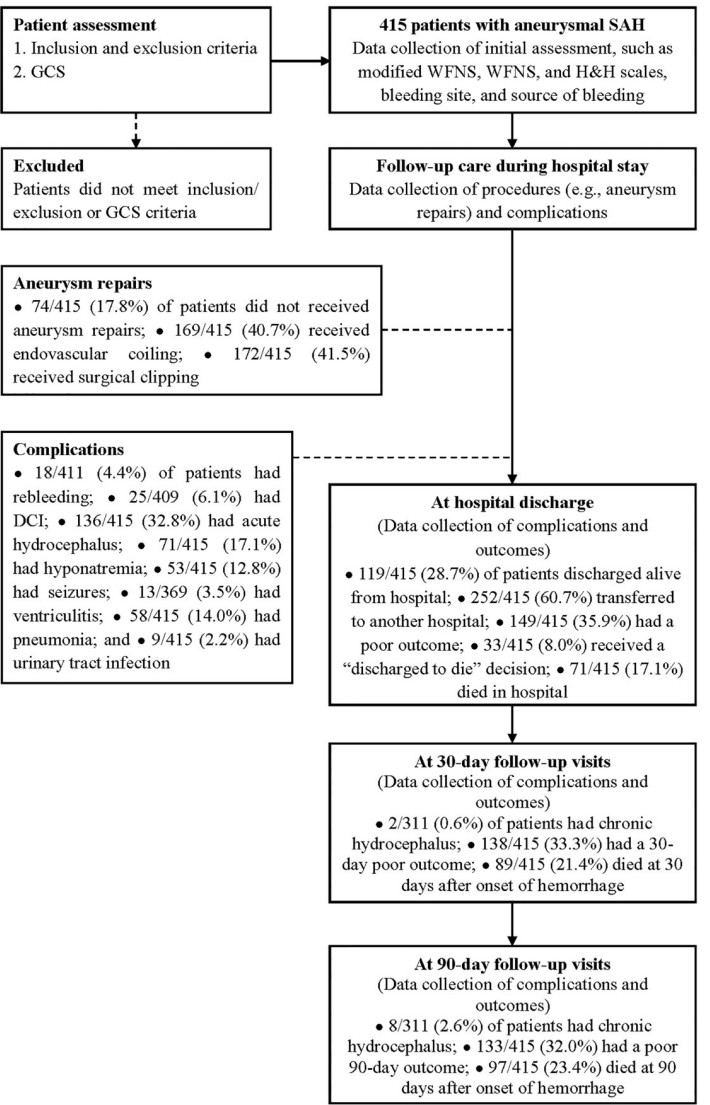

**Fig 1. Flowchart of the study design and assessment occasions.** (Abbreviations: "**discharged to die**", defined as patients were in grave condition or dying and were classified with a modified Rankin Scale score of 5 (severe disability) at the time of discharge; **DCI**: delayed cerebral ischemia; **GCS**: Glasgow Coma Scale; **H&H**: Hunt and Hess grading scale; **poor outcome**: defined as modified Rankin Scale scores of 4 (moderately severe disability) to 6 (death); **SAH**: subarachnoid hemorrhage; **WFNS**: World Federation of Neurosurgical Societies grading scale).

the poor outcome on day 30[th] after ictus, there were no significant differences between the AUROC curves among different test-pairwise, as shown in Table 2.

## Differences between the clinical outcomes of the adjacent grades

There were no significant differences in the mean 90-day mRS scores among all adjacent grades of the SAH grading scales except between grades IV and V (3.75 [SD: 2.46] vs 5.24 [SD: 1.68], p = 0.026, respectively) of the modified WFNS scale, between grades IV and V (3.75 [SD: 2.46] vs 5.24 [SD: 1.68], p = 0.026, respectively) of the WFNS scale, and between grades IV and V (2.96 [SD: 2.60] vs 4.97 [SD: 1.87], p<0.001, respectively) of the H&H scale (Table 3).

**Table 1. Baseline characteristics and outcomes of patients with aneurysmal subarachnoid haemorrhage.**

| Variables | N (%) |
|---|---|
| All cases | 415 |
| **Baseline characteristics** | |
| Age (year), median (IQR) | 57.0 (48.0–67.0) |
| Gender (male) | 198 (47.7) |
| Modified WFNS scale | |
| Grade I | 204 (49.2) |
| Grade II | 38 (9.2) |
| Grade III | 24 (5.8) |
| Grade IV | 99 (23.9) |
| Grade V | 50 (12.0) |
| WFNS scale | |
| Grade I | 204 (49.2) |
| Grade II | 48 (11.6) |
| Grade III | 14 (3.4) |
| Grade IV | 99 (23.9) |
| Grade V | 50 (12.0) |
| H&H scale | |
| Grade I | 45 (10.8) |
| Grade II | 168 (40.5) |
| Grade III | 62 (14.9) |
| Grade IV | 48 (11.6) |
| Grade V | 92 (22.2) |
| **Outcomes** | |
| Poor outcomes | |
| 30 days after ictus | 138 (33.3) |
| 90 days after ictus | 133 (32.0) |
| Deaths | |
| 30 days after ictus | 89 (21.4) |
| 90 days after ictus | 97 (23.4) |

Abbreviations: **H&H**, Hunt and Hess; **IQR**, interquartile range; **WFNS**, World Federation of Neurological Surgeons. See S2 to S9 Tables in S1 File for additional information.

Differences among the mean 30-day mRS scores of adjacent grades of the three SAH grading scales is available in Table 3.

## Associations between the grading scales and clinical outcomes

In the univariable logistic regression analyses, we found that most grades of the SAH grading scales, with grade I taken as the reference, were significantly associated with the increased risk of the 30- and 90-day poor outcomes, except for associations between grade II of the modified WFNS scale and the increased risk of the 30-day (OR: 1.475; 95% CI: 0.515–4.226; p = 0.469) and the 90-day poor outcome (OR: 1.566; 95% CI: 0.544–4.509; p = 0.406), and between grade II of the H&H scale and the increased risk of the 30-day (OR: 0.784; 95% CI: 0.269–2.287; p = 0.656) and the 90-day poor outcome (OR: 0.727; 95% CI: 0.247–2.139; p = 0.563) (Table 4).

When we added each SAH grading scale, with grade I taken as the reference, to the multi-variable logistic regression models, with the same set of confounding variables, for predicting

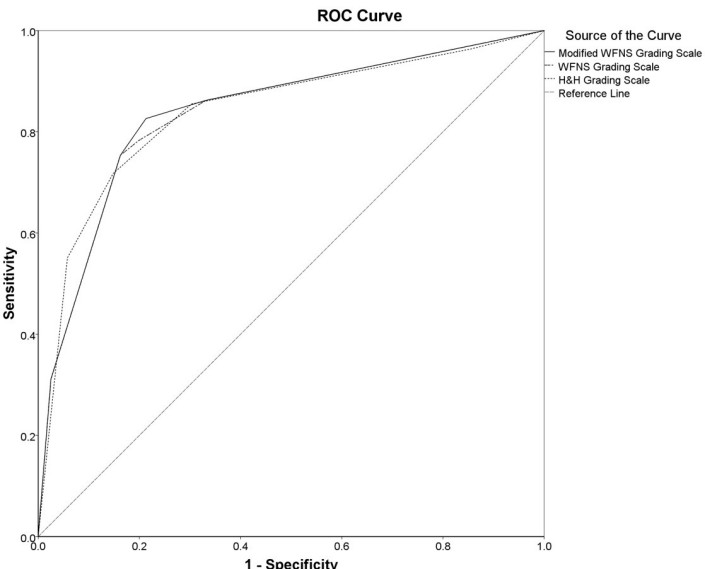

**Fig 2. The overall prognostic performance of the SAH grading scales for the poor outcomes on day 30th after ictus: The area under the ROC curves of the modified WFNS (AUROC: 0.839 [95% CI: 0.796–0.883]; cut-off value: ≥2.5; sensitivity: 82.6%; specificity: 78.7%; $P_{AUROC}$ <0.001), the WFNS (AUROC: 0.836 [95% CI: 0.793–0.880]; cut-off value: ≥3.5; sensitivity: 75.4%; specificity: 83.8%; $P_{AUROC}$ <0.001), and the H&H scales (AUROC: 0.839 (95% CI: 0.795–0.883); cut-off value: ≥3.5; sensitivity: 71.7%; specificity: 85.2%; $P_{AUROC}$ <0.001) for predicting the poor outcomes on day 30th after ictus in patients with aneurysmal SAH.** (Abbreviations: **AUROC**: areas under the receiver operating characteristic curve; **H&H**: Hunt and Hess; **poor outcome**: defined as a modified Rankin Scale [mRS] score of 4 to 6; **ROC**, receiver operating characteristic; **SAH**: subarachnoid hemorrhage; **WFNS**, World Federation of Neurological Surgeons).

the 30- and 90-day poor outcomes (S11 to S16 Tables in S1 File), we found that most grades of the modified WFNS, WFNS, and H&H scales were independently associated with the increased risk of the 90-day poor outcome, except for grade II (AOR: 0.990; 95% CI: 0.140–6.977; p = 0.992) of the modified WFNS scale, grades II (AOR: 2.097; 95% CI: 0.493–8.914; p = 0.316) and III (AOR: 6.198; 95% CI: 0.946–40.582; p = 0.057) of the WFNS scale, and grade II (AOR: 4.574; 95% CI: 0.683–30.654; p = 0.117) of the H&H scale (S14 to S16 Tables in S1 File). Associations between the grades of the modified WFNS, WFNS, and H&H scales and the risk of the 30-day poor outcome are available in S11 to S13 Tables in S1 File.

## The risk factors for poor outcomes

In the multivariable logistic regression models (Tables 5 and 6, S17 to S20 Tables in S1 File), with the same set of confounding variables, a modified WFNS grade of III to V (AOR: 9.090; 95% CI: 3.494–23.648; P<0.001) (Table 6) was more strongly associated with the increased risk of the 90-day poor outcome, compared to a WFNS grade of IV to V (AOR: 6.383; 95% CI: 2.661–15.310; P<0.001) (S19 Table in S1 File) and an H&H grade of IV to V (AOR: 6.146; 95% CI: 2.584–14.620; P<0.001) (S20 Table in S1 File). Similarly, we also found that a modified WFNS grade of III to V (AOR: 10.516; 95% CI: 4.092–27.026; P<0.001) (Table 5) was a more strongly independent predictor of the 30-day poor outcome, compared to a WFNS grade of IV to V (AOR: 6.879; 95% CI: 2.884–16.408; P<0.001) (S17 Table in S1 File) and an H&H grade of IV to V (AOR: 7.475; 95% CI: 3.202–17.449; P<0.001) (S18 Table in S1 File). Other factors were independently associated with the risk of the 30-day and 90-day poor outcome in the

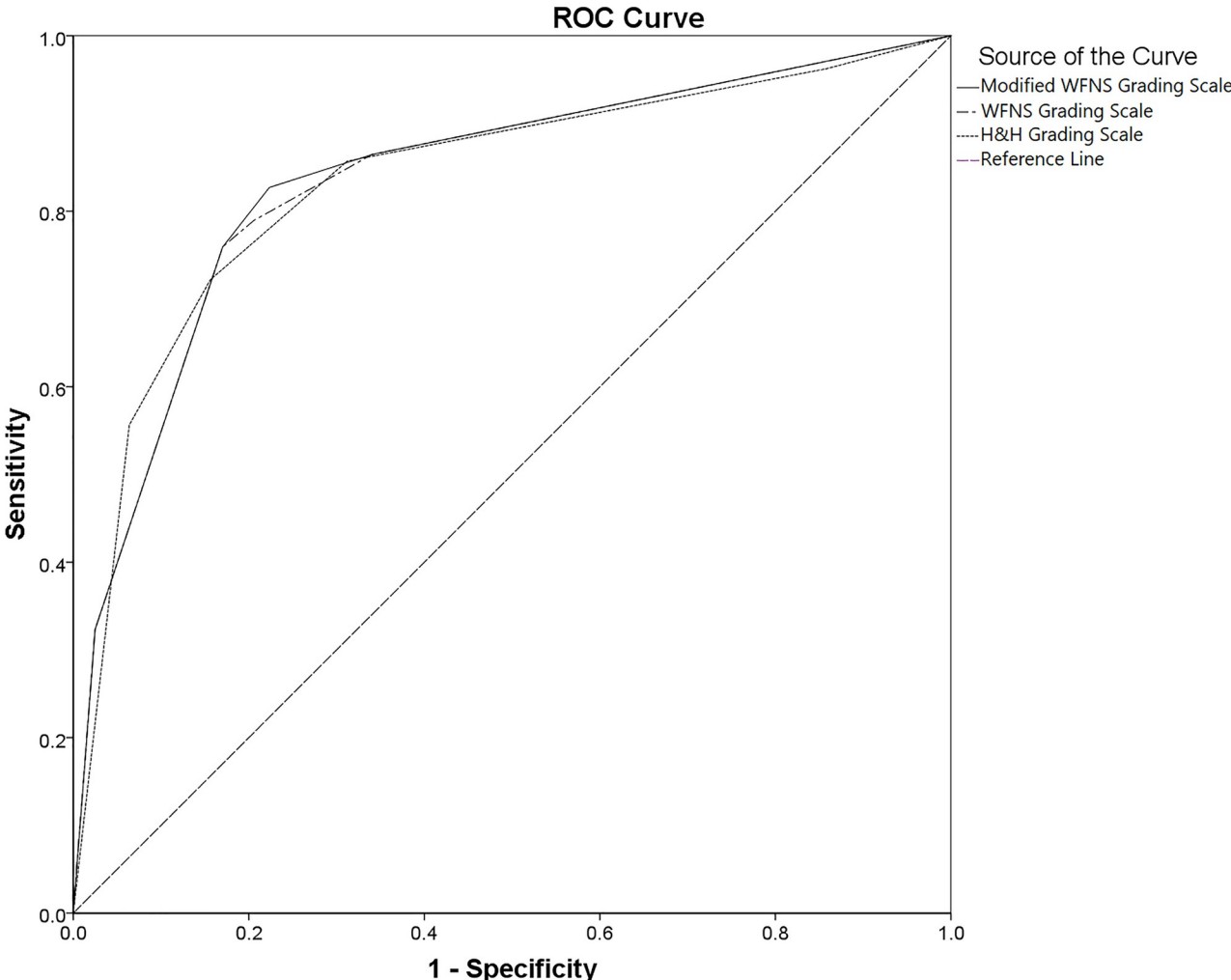

**Fig 3. The overall prognostic performance of the SAH grading scales for the poor outcomes on day 90ᵗʰ after ictus: The area under the ROC curves of the modified WFNS (AUROC: 0.839 [95% CI: 0.795–0.883]; cut-off value: ≥2.5; sensitivity: 82.7%; specificity: 77.7%; $P_{AUROC}$ <0.001), the WFNS (AUROC: 0.837 [95% CI: 0.793–0.881]; cut-off value: ≥3.5; sensitivity: 75.9%; specificity: 83.0%; $P_{AUROC}$ <0.001), and the H&H scales (AUROC: 0.836 [95% CI: 0.791–0.881]; cut-off value: ≥3.5; sensitivity: 72.2%; specificity: 84.4%; $P_{AUROC}$ <0.001) for predicting the poor outcomes on day 90ᵗʰ after ictus in patients with aneurysmal SAH.** (Abbreviations: **AUROC**: areas under the receiver operating characteristic curve; **H&H**: Hunt and Hess; **poor outcome**: defined as a modified Rankin Scale [mRS] score of 4 to 6; **ROC**, receiver operating characteristic; **SAH**: subarachnoid hemorrhage; **WFNS**, World Federation of Neurological Surgeons).

multivariable logistic regression models with different exposure variables are available in Tables 5 and 6 and S17 to S20 Tables in S1 File.

## Discussion

The present study revealed that nearly one-third of patients with aneurysmal SAH had poor outcomes on days 30ᵗʰ and 90ᵗʰ after ictus (33.3% and 32.0%, respectively), over one-fifth of whom died within 30 and 90 days of ictus (21.4% and 23.4%, respectively) (Fig 1 and Table 1). The modified WFNS, WFNS, and H&H scales all had good discriminatory ability concerning the prognosis of patients on days 30ᵗʰ and 90ᵗʰ after ictus (Figs 2 and 3, S10 Table in S1 File), with no significant differences between the AUROC curves among different test-pairwise (Table 2). There were only significant differences in the 30-day mean mRS scores between

**Table 2. Pairwise comparisons of AUROC of the modified WFNS, WFNS, and H&H scales for predicting the poor outcome (mRS of 4 to 6) after ictus in patients with aneurysmal SAH.**

| Comparison | AUROC difference (95% CI) | SE | Z -statistic | p-value |
|---|---|---|---|---|
| **Poor outcome on day 30[th] after ictus** | | | | |
| Modified WFNS and WFNS | 0.003 (0.000–0.006) | 0.0019 | 1.54 | 0.123 |
| Modified WFNS and H&H | 0.000(-0.018–0.018) | 0.0107 | 0.01 | 0.992 |
| WFNS and H&H | 0.003(-0.014–0.020) | 0.0104 | 0.27 | 0.787 |
| **Poor outcome on day 90[th] after ictus** | | | | |
| Modified WFNS and WFNS | 0.002(-0.001–0.005) | 0.0018 | 1.25 | 0.211 |
| Modified WFNS and H&H | 0.003(-0.015–0.021) | 0.0109 | 0.27 | 0.786 |
| WFNS and H&H | 0.001(-0.017–0.018) | 0.0106 | 0.07 | 0.946 |

Abbreviations: **AUC**, the area under the curve; **AUROC**, the area under the receiver operating characteristic; **H&H**, Hunt and Hess scale; **Modified WFNS**, Modified World Federation of Neurosurgical Societies scale; **SAH**, subarachnoid hemorrhage; **SE**, standard error; **WFNS**, World Federation of Neurosurgical Societies scale.

grades IV and V of the H&H scale and in the 90-day mean mRS scores between grades IV and V of the modified WFNS, the WFNS and the H&H scale (Table 3). In the univariable logistic regression analyses, with grade I taken as the reference, the modified WFNS scale did not show more gradual increases in OR for the 30- and 90-day poor outcome, in ascending grades,

**Table 3. Comparison of outcomes between the intergrades of the subarachnoid hemorrhage grading scales.**

| SAH grading scale | Day 30[th] after ictus | | | Day 90[th] after ictus | | |
|---|---|---|---|---|---|---|
| | mRS, mean (SD) | No. of patients | p-value[a] | mRS, mean (SD) | No. of patients | p-value[a] |
| **Modified WFNS scale** | | | | | | |
| I | 0.69 (1.50) | 204 | NA | 0.63 (1.55) | 204 | NA |
| II | 1.18 (2.02) | 38 | >0.999 | 1.00 (2.07) | 38 | >0.999 |
| III | 2.63 (2.50) | 24 | 0.372 | 2.29 (2.65) | 24 | 0.641 |
| IV | 3.86 (2.36) | 99 | 0.432 | 3.75 (2.46) | 99 | 0.073 |
| V | 5.26 (1.60) | 50 | 0.051 | 5.24 (1.68) | 50 | 0.026 |
| **WFNS scale** | | | | | | |
| I | 0.69 (1.50) | 204 | NA | 0.63 (1.55) | 204 | NA |
| II | 1.63 (2.29) | 48 | 0.096 | 1.40 (2.35) | 48 | 0.916 |
| III | 2.14 (2.41) | 14 | >0.999 | 1.86 (2.51) | 14 | >0.999 |
| IV | 3.86 (2.36) | 99 | 0.403 | 3.75 (2.46) | 99 | 0.159 |
| V | 5.26 (1.60) | 50 | 0.051 | 5.24 (1.68) | 50 | 0.026 |
| **H&H scale** | | | | | | |
| I | 0.98 (1.79) | 45 | NA | 0.87 (1.87) | 45 | NA |
| II | 0.65 (1.45)[b] | 168 | >0.999 | 0.59 (1.50)[b] | 168 | >0.999 |
| III | 2.10 (2.44) | 62 | 0.391 | 1.89 (2.49) | 62 | 0.588 |
| IV | 3.06 (2.47) | 48 | 0.306 | 2.96 (2.60) | 48 | 0.211 |
| V | 5.03 (1.76) | 92 | 0.001 | 4.97 (1.87) | 92 | <0.001 |

[a] Probability values were obtained by comparing the mean mRS score of a given grade with that of the mRS score just above it (nonparametric test by Dunn's multiple comparisons).

[b] The grade of the H&H scale shows the reversed rank order of the mean mRS scores.

Abbreviations

**H&H**: Hunt and Hess; **mRS**: modified Rankin Scale; **NA**: not applicable; **No.**: number; **SAH**: subarachnoid hemorrhage; **SD**: standard deviation; **WFNS**: World Federation of Neurological Surgeons.

**Table 4. Odds ratio for a poor outcome (mRS of 4 to 6) for the SAH grading scales.**

| SAH grading scale | Poor outcome on day 30th after ictus | | | | Poor outcome on day 90th after ictus | | | |
|---|---|---|---|---|---|---|---|---|
| | N | mRS: 4–6, no. (%) | OR (95% CI) | p-value | N | mRS: 4–6, no. (%) | OR (95% CI) | p-value |
| Modified WFNS scale | | | | | | | | |
| I | 204 | 19 (13.8) | reference | <0.001 | 204 | 18 (13.5) | reference | <0.001 |
| II | 38 | 5 (3.6) | 1.475 (0.515–4.226) | 0.469 | 38 | 5 (3.8) | 1.566 (0.544–4.509) | 0.406 |
| III | 24 | 10 (7.2) | 6.955 (2.720–17.784) | <0.001 | 24 | 9 (6.8) | 6.200 (2.380–16.154) | <0.001 |
| IV | 99 | 61 (44.2) | 15.630 (8.391–29.117) | <0.001 | 99 | 58 (43.6) | 14.618 (7.803–27.383) | <0.001 |
| V | 50 | 43 (31.2) | 59.812 (23.648–151.281) | <0.001 | 50 | 43 (32.3) | 63.476 (24.947–161.511) | <0.001 |
| WFNS scale | | | | | | | | |
| I | 204 | 19 (9.3) | reference | <0.001 | 204 | 18 (8.8) | reference | <0.001 |
| II | 48 | 11 (22.9) | 2.895 (1.272–6.587) | 0.011 | 48 | 10 (20.8) | 2.719 (1.164–6.350) | 0.021 |
| III | 14 | 4 (28.6) | 3.895 (1.114–13.621) | 0.033 | 14 | 4 (28.6) | 4.133 (1.177–14.520) | 0.027 |
| IV | 99 | 61 (61.6) | 15.630 (8.391–29.117) | <0.001 | 99 | 58 (58.6) | 14.618 (7.803–27.383) | <0.001 |
| V | 50 | 43 (86.0) | 59.812 (23.648–151.281) | <0.001 | 50 | 43 (86.0) | 63.476 (24.947–161.511) | <0.001 |
| H&H scale | | | | | | | | |
| I | 45 | 5 (11.1) | reference | <0.001 | 45 | 5 (11.1) | reference | <0.001 |
| II | 168 | 15 (8.9) | 0.784 (0.269–2.287) | 0.656 | 168 | 14 (8.3) | 0.727 (0.247–2.139) | 0.563 |
| III | 62 | 19 (30.6) | 3.535 (1.206–10.358) | 0.021 | 62 | 18 (29.0) | 3.273 (1.112–9.631) | 0.031 |
| IV | 48 | 23 (47.9) | 7.360 (2.478–21.860) | <0.001 | 48 | 22 (45.8) | 6.769 (2.277–20.121) | 0.001 |
| V | 92 | 76 (82.6) | 38.000 (12.973–111.306) | <0.001 | 92 | 74 (80.4) | 32.889 (11.362–95.201) | <0.001 |

Abbreviations

**CI**: confidence interval; **H&H**: Hunt and Hess; **mRS**: modified Rankin Scale; **N**: total number of patients for each grade; **no.**: number; **OR**: odds ratio; **SAH**: subarachnoid hemorrhage; **SD**: standard deviation; **WFNS**: World Federation of Neurological Surgeons.

See S11 to S16 Tables in S1 File for additional information.

compared to the WFNS and H&H scales (Table 4). In the multivariable logistic regression analyses, with the same set of confounding variables, the modified WFNS scale, however, showed more gradual increases in AOR for the 30- and 90-day poor outcome, in ascending grades, compared to the WFNS and H&H scales (S11 to S16 Tables in S1 File). Moreover, in the multivariable logistic regression analyses, with the same set of confounding variables, a modified WFNS grade of III to V (Tables 5 and 6) was more strongly associated with the increased risk of the 30- and 90-day poor outcome compared to a WFNS grade of IV to V (S17 to S18 in S1 File) and an H&H grade of IV to V (S19 to S20 in S1 File).

In our study, the rate of the 90-day poor outcome was lower than the rate reported in a previously published study (43.6%) [37]. The mortality rates of our patients on days 30th and 90th after ictus were also lower than rates reported in previous studies (22–25% and 25–29%, respectively) [38, 39]. These findings might be due to several advances in the past decades in the medical diagnosis and care of patients with aneurysmal SAH, including medical advances, new systems of standardized care in neurocritical care units, and new surgical and endovascular techniques for aneurysmal SAH [4, 40]. Additionally, as unique to patients with aneurysmal SAH in Vietnam, many are not transferred to a central hospital and are relegated to death in the local hospitals as well as dying outside of the hospital [41]. Therefore, these particular differences in poor outcomes or deaths might be accounted for because our cohort is likely to be highly selected. These differences might also be attributed to our study only including patients presenting to the participating hospitals within four days after ictus and excluding patients for whom the admission GCS was unable to be scored (e.g., patients intubated and under sedation

**Table 5. Factors associated with the 30-day poor outcome (mRS of 4 to 6) in patients with aneurysmal subarachnoid hemorrhage.**

| Factors | Univariable logistic regression analyses[a] | | | | Multivariable logistic regression analysis[b] | | | |
|---|---|---|---|---|---|---|---|---|
| | OR | 95% CI for OR | | p-value | AOR | 95% CI for AOR | | p-value |
| | | Lower | Upper | | | Lower | Upper | |
| **Demographics** | | | | | | | | |
| Age $\geq$ 60 years | 2.407 | 1.586 | 3.654 | <0.001 | 2.921 | 1.281 | 6.661 | 0.011 |
| **Risk factors of aneurysmal SAH** | | | | | | | | |
| Hypertension | 2.056 | 1.354 | 3.122 | 0.001 | NA | NA | NA | NA |
| **Comorbidities** | | | | | | | | |
| Diabetes mellitus | 3.751 | 1.669 | 8.433 | 0.001 | NA | NA | NA | NA |
| **Neuroimaging findings on admission** | | | | | | | | |
| Location of blood within the subarachnoid space: | | | | | | | | |
| Basal cistern | 2.685 | 1.728 | 4.173 | <0.001 | NA | NA | NA | NA |
| Sylvian fissure | 2.981 | 1.125 | 7.900 | 0.028 | NA | NA | NA | NA |
| Interhemispheric fissure | 1.960 | 1.207 | 3.183 | 0.007 | NA | NA | NA | NA |
| Interpeduncular fossa | 3.254 | 1.995 | 5.308 | <0.001 | 2.559 | 1.070 | 6.122 | 0.035 |
| Suprasellar cistern | 1.929 | 1.221 | 3.047 | 0.005 | NA | NA | NA | NA |
| Ambient cistern | 3.054 | 1.902 | 4.906 | <0.001 | NA | NA | NA | NA |
| Quadrigeminal cistern | 5.141 | 3.270 | 8.085 | <0.001 | NA | NA | NA | NA |
| IVH | 3.013 | 1.847 | 4.914 | <0.001 | NA | NA | NA | NA |
| ICH | 1.860 | 1.142 | 3.029 | 0.013 | NA | NA | NA | NA |
| Aneurysm locations | | | | | | | | |
| PCoA aneurysm | 0.554 | 0.299 | 1.026 | 0.061 | NA | NA | NA | NA |
| VA aneurysm | 3.341 | 1.265 | 8.820 | 0.015 | NA | NA | NA | NA |
| **Severity of aneurysmal SAH on admission** | | | | | | | | |
| Modified WFNS grade of III to V[c] | 17.551 | 10.374 | 29.693 | <0.001 | 10.516 | 4.092 | 27.026 | <0.001 |
| **Aneurysm repairs and other treatments** | | | | | | | | |
| Aneurysm repairs: | | | | | | | | |
| No aneurysm repair | Reference | | | <0.001 | Reference | | | <0.001 |
| Endovascular coiling | 0.009 | 0.003 | 0.027 | <0.001 | 0.009 | 0.002 | 0.032 | <0.001 |
| Surgical clipping | 0.020 | 0.007 | 0.059 | <0.001 | 0.011 | 0.003 | 0.040 | <0.001 |
| Surgical hematoma evacuation[d] | 2.438 | 1.297 | 4.583 | 0.006 | NA | NA | NA | NA |
| EVD[e] | 3.537 | 1.846 | 6.777 | <0.001 | 4.594 | 1.595 | 13.228 | 0.005 |
| Nimodipine | 0.076 | 0.028 | 0202 | <0.001 | NA | NA | NA | NA |
| **Complications** | | | | | | | | |
| Rebleeding | 7.868 | 2.537 | 24.396 | <0.001 | 19.734 | 3.879 | 100.399 | <0.001 |
| DCI | 7.316 | 2.848 | 18.792 | <0.001 | 20.914 | 5.331 | 82.049 | <0.001 |
| Acute hydrocephalus | 3.134 | 2.034 | 4.830 | <0.001 | NA | NA | NA | NA |
| Pneumonia | 4.091 | 2.295 | 7.292 | <0.001 | 2.826 | 1.122 | 7.119 | 0.028 |

(*Continued*)

**Table 5.** (Continued)

| Factors | Univariable logistic regression analyses[a] | | | Multivariable logistic regression analysis[b] | | | |
|---|---|---|---|---|---|---|---|
| | OR | 95% CI for OR | | p-value | AOR | 95% CI for AOR | | p-value |
| | | Lower | Upper | | | Lower | Upper | |
| Constant | | | | | 0.741 | | | 0.681 |

[a]Each variable of the demographics, risk factors for aneurysmal SAH, comorbidities, initial clinical, neuroimaging and laboratory characteristics, the severity of aneurysmal SAH (i.e., modified WFNS scale) on admission, treatments, and complications was analyzed in the univariable logistic regression model and was considered in the multivariable logistic regression model if the P-value was <0.05 in univariable logistic regression analysis, as well as clinically crucial factors.

[b]All selected variables were included in the multivariable logistic regression model with the stepwise backward elimination method. Variables, then, were deleted stepwise from the full model until all remaining variables were independently associated with poor outcomes.

[c] The grades of the modified WFNS scale which were higher than or equal to the cut-off value.

[d] Surgical hematoma evacuation was defined as any surgical procedure evacuating epidural, subdural, intraventricular, or intraparenchymal hematoma, such as decompressive craniotomy, open craniotomy, or minimally invasive surgery.

[e] The reason for the EVD insertion was the complication of acute hydrocephalus and others (See S5 and S9 Tables in S1 File for additional information).

Abbreviations: **AOR**, adjusted odds ratio; **CI**, confidence interval; **DCI**, delayed cerebral ischemia; **EVD**: external ventricular drainage; **ICH**, intracerebral hemorrhage; **IVH**, intraventricular hemorrhage; **mRS**, modified Rankin Scale; **NA**, not available; **OR**, odds ratio; **PCoA**, posterior communicating artery; **SAH**, subarachnoid hemorrhage; **VA**, vertebral artery; **WFNS**, World Federation of Neurosurgical Societies.

See S17 to S20 Tables in S1 File for additional information.

before arrival at the central hospital). Thus, these factors have resulted in an implicit selection bias and an enrolment and inclusion incompletion of patients in the study database. As a result, our cohort is likely to be underestimated in the poor outcome and mortality rates.

Although there were no significant differences between the AUROC curves among different test-pairwise, our modified WFNS, WFNS, and H&H scales all had good discriminatory ability concerning the prognosis of patients on days 30th and 90th after ictus. However, our study showed only significant differences in the 30-day mean mRS scores between grades IV and V of the H&H scale and in the 90-day mean mRS scores between grades IV and V of the modified WFNS, the WFNS and the H&H scale. To date, there is no universally accepted scale to assess the clinical condition of these patients at the time of admission [7, 42]. Although the WFNS and H&H scales are both widely used in clinical practice and research reports, the lack of formal validation of the WFNS scale might lead to occasional overlap between grades [11, 14, 15], and the interobserver agreement for the H&H scale is poor [8, 9]. The WFNS scale has two main advantages over the GCS alone. It compresses the GCS into five grades, which may create greater intergrade differences in outcome. It includes the presence of a focal motor deficit axis. However, the amount of additional prognostic power derived from adding this axis is unknown [7]. Our study showed no significant differences in the 30- and 90-day outcomes (mean mRS scores) among all adjacent grades of the WFNS scale except for the significant differences in the 90-day mean mRS scores between grades IV and V. A previous study also showed that the differences in the outcomes between grades II and III failed to reach statistical significance on the WFNS scale [15]. These findings might be due to the lack of formal validation of the WFNS scale, which might lead to occasional overlap between grades (particularly between grades II and III), where the outcomes predicted by the assigned grade may not differ substantially [11, 14]. The present study also showed no significant differences in the 30- and 90-day outcomes (mean mRS scores) between the adjacent grades of the H&H scale except for between grades IV and V. Although the H&H scale is easy to administer, the classifications are arbitrary, some of the terms are vague (e.g., drowsy, stupor, and deep coma), and some patients may present with initial features that defy placement within a single grade [7]. As an

**Table 6. Factors associated with the 90-day poor outcome (mRS of 4 to 6) in patients with aneurysmal subarachnoid hemorrhage.**

| Factors | Univariable logistic regression analyses[a] | | | | Multivariable logistic regression analysis[b] | | | |
|---|---|---|---|---|---|---|---|---|
| | OR | 95% CI for OR | | p-value | AOR | 95% CI for AOR | | p-value |
| | | Lower | Upper | | | Lower | Upper | |
| **Demographics** | | | | | | | | |
| Age ≥ 60 years | 2.581 | 1.691 | 3.939 | <0.001 | 3.554 | 1.524 | 8.287 | 0.003 |
| **Risk factors of aneurysmal SAH** | | | | | | | | |
| Hypertension | 2.133 | 1.400 | 3.250 | <0.001 | NA | NA | NA | NA |
| **Comorbidities** | | | | | | | | |
| Diabetes mellitus | 3.986 | 1.772 | 8.968 | 0.001 | NA | NA | NA | NA |
| **Neuroimaging findings on admission** | | | | | | | | |
| Location of blood within the subarachnoid space: | | | | | | | | |
| Basal cistern | 2.718 | 1.738 | 4.251 | <0.001 | NA | NA | NA | NA |
| Sylvian fissure | 2.811 | 1.060 | 7.454 | 0.038 | NA | NA | NA | NA |
| Interhemispheric fissure | 2.063 | 1.257 | 3.385 | 0.004 | NA | NA | NA | NA |
| Interpeduncular fossa | 3.206 | 1.953 | 5.264 | <0.001 | NA | NA | NA | NA |
| Suprasellar cistern | 1.882 | 1.186 | 2.986 | 0.007 | NA | NA | NA | NA |
| Ambient cistern | 2.990 | 1.852 | 4.829 | <0.001 | NA | NA | NA | NA |
| Quadrigeminal cistern | 4.414 | 3.431 | 8.545 | <0.001 | 2.665 | 1.139 | 6.234 | 0.024 |
| IVH | 2.793 | 1.711 | 4.559 | <0.001 | NA | NA | NA | NA |
| ICH | 1.879 | 1.151 | 3.068 | 0.012 | NA | NA | NA | NA |
| Aneurysm locations | | | | | | | | |
| PCoA aneurysm | 0.590 | 0.318 | 1.094 | 0.094 | NA | NA | NA | NA |
| VA aneurysm | 3.542 | 1.341 | 9.356 | 0.011 | NA | NA | NA | NA |
| **Severity of aneurysmal SAH on admission** | | | | | | | | |
| Modified WFNS grade of III to V[c] | 16.625 | 9.790 | 28.233 | <0.001 | 9.090 | 3.494 | 23.648 | <0.001 |
| **Aneurysm repairs and other treatments** | | | | | | | | |
| Aneurysm repairs: | | | | | | | | |
| No aneurysm repair | Reference | | | <0.001 | Reference | | | <0.001 |
| Endovascular coiling | 0.011 | 0.004 | 0.030 | <0.001 | 0.009 | 0.002 | 0.032 | <0.001 |
| Surgical clipping | 0.023 | 0.009 | 0.062 | <0.001 | 0.014 | 0.004 | 0.049 | <0.001 |
| Surgical hematoma evacuation[d] | 2.111 | 1.123 | 3.971 | 0.020 | NA | NA | NA | NA |
| EVD[e] | 3.382 | 1.773 | 6.453 | <0.001 | 5.365 | 1.852 | 15.544 | 0.002 |
| Nimodipine | 0.071 | 0.027 | 0.191 | <0.001 | NA | NA | NA | NA |
| **Complications** | | | | | | | | |
| Rebleeding | 6.133 | 2.138 | 17.594 | 0.001 | 22.913 | 4.560 | 115.135 | <0.001 |
| DCI | 6.245 | 2.538 | 15.365 | <0.001 | 16.210 | 4.279 | 61.412 | <0.001 |
| Acute hydrocephalus | 3.134 | 2.028 | 4.842 | <0.001 | NA | NA | NA | NA |
| Pneumonia | 4.022 | 2.263 | 7.148 | <0.001 | 3.375 | 1.306 | 8.722 | 0.012 |

(*Continued*)

**Table 6.** (Continued)

| Factors | Univariable logistic regression analyses[a] | | | Multivariable logistic regression analysis[b] | | | |
|---|---|---|---|---|---|---|---|
| | OR | 95% CI for OR | | p-value | AOR | 95% CI for AOR | | p-value |
| | | Lower | Upper | | | Lower | Upper | |
| Constant | | | | | 0.731 | | | 0.631 |

[a]Each variable of the demographics, risk factors for aneurysmal SAH, comorbidities, initial clinical, neuroimaging and laboratory characteristics, the severity of aneurysmal SAH (i.e., modified WFNS scale) on admission, treatments, and complications was analyzed in the univariable logistic regression model and was considered in the multivariable logistic regression model if the P-value was <0.05 in univariable logistic regression analysis, as well as clinically crucial factors.

[b]All selected variables were included in the multivariable logistic regression model with the stepwise backward elimination method. Variables, then, were deleted stepwise from the full model until all remaining variables were independently associated with poor outcomes.

[c] The grades of the modified WFNS scale which were higher than or equal to the cut-off value.

[d] Surgical hematoma evacuation was defined as any surgical procedure evacuating epidural, subdural, intraventricular, or intraparenchymal hematoma, such as decompressive craniotomy, open craniotomy, or minimally invasive surgery.

[e] The reason for the EVD insertion was the complication of acute hydrocephalus and others (See S5 and S9 Tables in S1 File for additional information).

Abbreviations: **AOR**, adjusted odds ratio; **CI**, confidence interval; **DCI**, delayed cerebral ischemia; **EVD**: external ventricular drainage; **ICH**, intracerebral hemorrhage; **IVH**, intraventricular hemorrhage; **mRS**, modified Rankin Scale; **NA**, not available; **OR**, odds ratio; **PCoA**, posterior communicating artery; **SAH**, subarachnoid hemorrhage; **VA**, vertebral artery; **WFNS**, World Federation of Neurosurgical Societies.

See S17 to S20 Tables in S1 File for additional information.

example, a rare presentation of SAH may include severe headache (i.e., grade II), normal level of consciousness, and severe hemiparesis (i.e., grade IV). In such cases, the clinician must subjectively decide which of the presenting features is the most important for determining the grade. Therefore, our findings might be accounted for by the poor interobserver agreement and might also contribute to the conflicting data regarding the utility of the H&H scale for prognosis [8, 9, 11]. The modified WFNS scale is very easy to apply and is based solely on the GCS, which has better outcome predictability [15, 26, 27]. Unlike the originally suggested modified WFNS scale, for which the 90-day mean mRS scores of each grade differs from that of adjacent grades with a statistical significance except between grades III and IV [15], the present study showed a significant difference in the 90-day mean mRS scores only between grades IV and V of the modified WFNS scale. This variation might be because of the differences concerning the time points of GCS evaluation as well as assessment of SAH grading scale (i.e., after initial stabilization vs upon admission) between the two studies. A retrospective cohort study has shown that the outcome after aneurysmal SAH was best predicted by assessing WFNS grade after neurocritical stabilization and neurosurgical/neuro-interventional procedures [43]. The fact that significant differences in the 30- and 90-day mean mRS scores between the adjacent grades were not observed more clearly in the modified WFNS scale compared to those in the WFNS and the H&H scales in our study suggests that the modified WFNS scale maybe not preferable to the WFNS and H&H scales.

In our univariable logistic regression analyses, with grade I taken as the reference, the modified WFNS scale did not show more gradual increases in OR for the 30- and 90-day poor outcome, in ascending grades, compared to the WFNS and H&H scales. When we added each SAH grading scale (i.e., modified WFNS, WFNS, or H&H scale) to the multivariable logistic regression model, with the same set of confounding variables, for predicting the 30- and 90-day poor outcome, we found a more gradual increase in AOR of the modified WFNS scale, in ascending grades, compared to those of the WFNS and H&H scales, which might be due to the same or more number of grades, that was independently associated with the increased risk of poor outcome, was observed in the modified WFNS

scale compared to the WFNS and the H&H scale. Previous critiques identified a caution problem with ORs [44–48], and a recent literature review has raised this issue again [49]; namely, (i) there is no single OR; instead, any estimated OR is conditional on the data and the model specification; (ii) ORs should not be compared across different studies using different samples from different populations; and (iii) nor should they be compared across models with different sets of confounding variables [49]. However, we used the univariable logistic regression analyses, with the same grade (i.e., I) taken as the reference, and the multivariable logistic regression analyses, with the same set of confounding variables, to determine the relationships among the grades on the modified WFNS, WFNS, and H&H scales and the actual outcomes. Therefore, a more gradual increase in effect size (i.e., AOR) of the modified WFNS scale, in ascending grades, for predicting the poor outcome in our multivariable logistic regression analyses suggests that the modified WFNS may be preferable to the WFNS and H&H scales.

Although the advances in diagnostic and treatment strategies for aneurysmal SAH have substantially improved the outcomes of hospitalized patients in recent decades [50–53]. predicting the outcome of aneurysmal SAH remains a problematic issue. The clinical condition can vary during the acute phase, and complications occurring during the clinical course and treatments rendered can influence the outcome [28, 54, 55]. In the present study, complications (e.g., rebleeding, DCI, pneumonia) also accounted for a substantial proportion of patients with aneurysmal SAH and contributed significantly to a high rate of poor outcomes. Nevertheless, grading patients with SAH on admission is crucial for clinical and research purposes. Most grading systems are used in practice to standardize the clinical classification of patients with SAH based only on the initial neurologic examination and the appearance of blood on the initial head CT [5, 6, 25, 56]. Therefore, a scale applied upon admission will never give a 100% perfect prediction for the outcome. Although our modified WFNS, WFNS, and H&H scales all had good discriminatory ability concerning the prognosis of patients on days 30th and 90th after ictus, there were no significant differences between the AUROC curves among different test-pairwise. However, a modified WFNS grade of III to V, a WFNS grade of IV to V and an H&H grade of IV to V were independent predictors of the 30- and 90-day poor outcome. Because of the better effect size (i.e., AOR) of modified WFNS grade of III to V for predicting the 30- and 90-day poor outcome, the modified WFNS scale was preferable to the WFNS and the H&H scales.

An advantage of the present study was data from the multicenter, which had little missing data (S21 Table in S1 File). However, the present study has some limitations as follows: Firstly, our data are from a selected population of cases that were mainly transferred to the three highest-level public sector hospitals in Vietnam. Therefore, the number of patients with aneurysmal SAH is likely to be considerably higher. Secondly, there is a lack of interobserver agreement, since we were unable to provide multiple clinicians for each of the patients' initial evaluations. Finally, this study only included patients presenting to the participating hospitals within 4 days of ictus and excluded patients for whom admission GCS was unable to be scored (e.g., patients intubated and under sedation before arrival at the central hospital). These factors might have resulted in an implicit selection bias and an incomplete enrolment and inclusion of patients in the database. Differences in figures found between Vietnam and other nations might be accounted for by these factors above.

## Conclusions

This study investigated a selected cohort of patients with aneurysmal SAH, a high rate of poor outcomes and a high mortality rate presented to central hospitals in Vietnam. The modified

WFNS, WFNS, and H&H scales all had good discriminatory abilities for the prognosis of patients with aneurysmal SAH. Because of the better effect size in predicting poor outcomes, the modified WFNS scale seems preferable to the WFNS and H&H scales.

## Supporting information

**S1 File.**
(PDF)

## Acknowledgments

We thank all staff of the Center for Emergency Medicine, the Stroke Center, the Department of Neurosurgery, and the Radiology Centre at Bach Mai Hospital for their support with this study. We thank all staff of the Neurosurgery Center at Vietnam-Germany Friendship Hospital for their support with this study, as well as the Emergency and Critical Care Department at Hanoi Medical University Hospital. We also thank the staff of the Faculty of Public Health at Thai Binh University of Medicine and Pharmacy for their support and statistical advice. Finally, we thank Miss Truc-Cam Nguyen from Stanford University, Stanford, California, the United States of America, and Miss Mai Phuong Nguyen from the Hotchkiss School, Lakeville, Connecticut, the United States of America, for their support of our manuscript.

## Author Contributions

**Conceptualization:** Chinh Quoc Luong.

**Data curation:** Tuan Anh Nguyen, Ton Duy Mai, Luu Dang Vu, Co Xuan Dao, Hung Manh Ngo, Hai Bui Hoang, Tuan Anh Tran, Trang Quynh Pham, Linh Quoc Nguyen, Phuong Viet Dao, Duong Ngoc Nguyen, Hien Thi Thu Vuong, Hung Dinh Vu, Dong Duc Nguyen, Thanh Dang Vu, Dung Tien Nguyen, Anh Le Ngoc Do, Chinh Quoc Luong.

**Formal analysis:** Dung Thi Pham, My Ha Nguyen, Chinh Quoc Luong.

**Investigation:** Tuan Anh Nguyen, Ton Duy Mai, Luu Dang Vu, Co Xuan Dao, Hung Manh Ngo, Hai Bui Hoang, Tuan Anh Tran, Trang Quynh Pham, Dung Thi Pham, My Ha Nguyen, Linh Quoc Nguyen, Phuong Viet Dao, Duong Ngoc Nguyen, Hien Thi Thu Vuong, Hung Dinh Vu, Dong Duc Nguyen, Thanh Dang Vu, Dung Tien Nguyen, Anh Le Ngoc Do, Cuong Duy Nguyen, Son Ngoc Do, Hao The Nguyen, Chi Van Nguyen, Anh Dat Nguyen, Chinh Quoc Luong.

**Methodology:** Tuan Anh Nguyen, Ton Duy Mai, Luu Dang Vu, Co Xuan Dao, Hung Manh Ngo, Hai Bui Hoang, Tuan Anh Tran, Trang Quynh Pham, Dung Thi Pham, My Ha Nguyen, Chinh Quoc Luong.

**Project administration:** Chinh Quoc Luong.

**Supervision:** Co Xuan Dao, Son Ngoc Do, Chinh Quoc Luong.

**Writing – original draft:** Chinh Quoc Luong.

**Writing – review & editing:** Tuan Anh Nguyen, Ton Duy Mai, Luu Dang Vu, Co Xuan Dao, Cuong Duy Nguyen, Son Ngoc Do, Hao The Nguyen, Chi Van Nguyen, Anh Dat Nguyen, Chinh Quoc Luong.

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
