## [Decision Letter · Decision Letter 0]

31 Jan 2023

PONE-D-22-32313Validation of the accuracy of the modified World Federation of Neurosurgical Societies subarachnoid haemorrhage grading scale for predicting the outcomes of patients with aneurysmal subarachnoid haemorrhagePLOS ONE

Dear Dr. Luong,

Thank you for submitting your manuscript to PLOS ONE. After careful consideration, we feel that it has merit but does not fully meet PLOS ONE’s publication criteria as it currently stands. Therefore, we invite you to submit a revised version of the manuscript that addresses the points raised during the review process.

We look forward to receiving your revised manuscript.

Kind regards,

Martin Kieninger

Academic Editor

PLOS ONE

Journal Requirements:

2. We noted in your submission details that a portion of your manuscript may have been presented or published elsewhere. Please clarify whether this conference proceeding or publication was peer-reviewed and formally published. If this work was previously peer-reviewed and published, in the cover letter please provide the reason that this work does not constitute dual publication and should be included in the current manuscript.

3. Please include a caption for figure 2.

4. Please upload a copy of Figure 2, to which you refer in your text on page 20. If the figure is no longer to be included as part of the submission please remove all reference to it within the text.

Additional Editor Comments:

I would like to apologize again for the long wait. However, the diametrically different assessments of Reviewer 1 and Reviewer 2 made a third review necessary.

Reviewers' comments:

Reviewer's Responses to Questions

**Comments to the Author**

1. Is the manuscript technically sound, and do the data support the conclusions?

Reviewer #1: Yes

Reviewer #2: Yes

Reviewer #3: Yes

2. Has the statistical analysis been performed appropriately and rigorously? 

Reviewer #1: Yes

Reviewer #2: I Don't Know

Reviewer #3: Yes

3. Have the authors made all data underlying the findings in their manuscript fully available?

Reviewer #1: Yes

Reviewer #2: Yes

Reviewer #3: Yes

4. Is the manuscript presented in an intelligible fashion and written in standard English?

Reviewer #1: Yes

Reviewer #2: Yes

Reviewer #3: Yes

5. Review Comments to the Author

Reviewer #1: This is a very interesting study concerning the validation of modified World Federation of Neurosurgical Societies (WFNS) subarachnoid haemorrhage grading scale in predicting the 90-day poor outcome defined as modified Rankin Scale scores of 4 to 6. The validation process was also correlated to WFNS and H&H scores. The result showed more accurate than those in WFNS and H&H scores. The manuscript is well written and should be published.

Reviewer #2: Thank you for submitting your manuscript.

Your multi center analysis of the 3 SAH grades (WFNS, mWFNS and Hunt and Hess) was thoroughly performed.

I however do have some comments.

1. It is stated that the WFNS is more complex to administer than the H&H scales because it requires the GCS and the motor function. First, I am not sure how it is more complicated than the H&H, since the GCS is a standard validated objective score that all neurosurgeon, and even emergency medicine doctors are all comfortable with. The H&H grade is very subjective. In addition, the WFNS score requires the GCS and the presence of a motor deficit only, which isn’t to complicated.

2. How many patients were excluded because of the initial GCs was not scored? Looking at the chart of the referring hospital or even the ambulance chart would yield the GCS.

3. Why were patients lost at 90 day? If surgical or even treated by embolization, were follow ups not performed?

4. The management of the aneurysmal SAH, let it be surgical or endovascular treatment, if very variable. The treatment option may play an important role in the outcome of the patient. Not having a standardized treatment option in these patients, and looking at the outcome is not adequate. This is a main caveat of the study.

5. It is stated that 415 patients presented to the study sites. How many were excluded because of the inadequate documentation of GCS? And how many and why where they lost to follow-up?

6. The conclusion that the mWFSN score was strongest associated with an increased risk of poor outcome at 30/90 days of ictus is a strong statement. This needs to be revised accordingly to the limits of the study.

7. Although their data do support their conclusion, I do not believe that this has any clinical impact, nor does it help in the clinical decision making of the surgeon /radiologist.

Although interesting, the data inclusion criteria is very weak.

Reviewer #3: The authors present the results of a prospective trial in patients with aneurysmal SAH, attempting to analyze the predictive value of SAH scales for poor outcome. This is an interesting study that carries high importance for the field. However, there are a number of aspects that need to be addressed before publication.

• How long was the median follow-up time?

• Were there any changes of patient status after 90 days ?

• What was the outcome at discharge?

• In the participants and treatment section, it is mentioned that “In the case of aphasia, patients were classified according to the clinically possible GCS scores derived from their eye and motor scores. How exactly is the “eye score” performed, and how was it implemented in the GS scale?

• How were missing data handled?

• Since the applied SAH grading systems are on an ordinal scale niveau, how meaningful is the approach of calculating ROC, which usually requires continuous data as the diagnostic input variable? For example, formulating a resulting cut-off value of 2.5 for poor outcomes in the modified WFNS scale is clinically challenging to implement. Please elaborate.

• Comparing odd`s ratios between groups and models has been criticized, in fact, this approach has recently been rejected as an adequate method in this context. Several authors have argued that odds ratios will change if variables are added to the model, even if those additional variables are independent from the other variables. This concern particularly applies to the analysis of differences between clinical outcomes of the adjacent grades. Please comment on this critical aspect.

• Clinically, there are a number of variables that may influence the results of this analysis. In particular, was there an influence of aneurysm location and the frequency of vasospasm / delayed ischemia? How many patients received decompressive craniectomy?

• The strongest risk factor for poor outcome was the modified WFNS score of 3-5, which translates into an initial GCS score of 3-13. This variability makes the application of such a “risk - factor” somewhat challenging to implement in the clinical setting.

• To my best knowledge, the modified WFNS score appears to carry a better discriminatory value for good outcomes compared to the original WFNS score. Do the authors find similar results for the patients with a good outcome?

• Since the initial SAH grading is influenced by acute hydrocephalus, how many patients with acute hydrocephalus improved after implantation of an intraventricular drain? Did these patients show a different result regarding the SAH scales ad their prediction of poor outcomes?

6. PLOS authors have the option to publish the peer review history of their article (what does this mean?). If published, this will include your full peer review and any attached files.

Reviewer #1: No

Reviewer #2: No

Reviewer #3: No

---

## [Author Response · Author response to Decision Letter 0]

31 Mar 2023

Professor Martin Kieninger

Academic Editor

PLOS ONE

March 17, 2023

Dear Prof. Martin Kieninger,

On behalf of all authors, I am resubmitting herewith our revised manuscript entitled “Validation of the accuracy of the modified World Federation of Neurosurgical Societies subarachnoid haemorrhage grading scale for predicting the outcomes of patients with aneurysmal subarachnoid haemorrhage” (PONE-D-22-32313R1).

We sincerely appreciate the kind comments and points raised by the Editors and by the Reviewers. We have carefully considered all comments and suggestions and revised our manuscript following each of these points. These comments have enabled us to substantially improve our manuscript. We hope that Editor will find our revised manuscript suitable for publication in PLOS ONE.

We confirm that this work is original and has not been published elsewhere nor is it currently under consideration for publication elsewhere. All authors have read, approved the manuscript, and agreed to authorship and order of authorship for this manuscript, and all authors have the appropriate permissions and rights to the reported data.

We have provided our point-by-point responses to the comments of the Editors and the Reviewers attached.

We thank you for your kind consideration of this submission.

Sincerely yours,

Chinh Quoc Luong, MD., PhD.

Center for Emergency Medicine,

Bach Mai Hospital,

No. 78, Giai Phong, Phuong Mai ward, Dong Da district, Hanoi 100000, Vietnam

Email: luongquocchinh@gmail.com

We thank the Editors and the Reviewers for the valuable comments and suggestions that greatly helped us to improve the contents of this paper. In what follows, we will use the boldface to indicate comments from the Editors and the Reviewers, the standard font face for our responses and we highlighted in yellow the modifications that we did to the manuscript.

RESPONSE TO EDITORS

Thank you for submitting your manuscript to PLOS ONE. After careful consideration, we feel that it has merit but does not fully meet PLOS ONE’s publication criteria as it currently stands. Therefore, we invite you to submit a revised version of the manuscript that addresses the points raised during the review process.

Our answer:

We thank you for the positive feedback. We have carefully considered the Reviewers' comments and suggestions and have revised our manuscript following each of these points.

Thank you for this comment. We have submitted the revised manuscript on time.

Our answer:

Thank you for this comment. We have included a rebuttal letter, a marked-up copy of the manuscript, and an unmarked version of the revised manuscript when submitting our revised manuscript.

Our answer:

Thank you for this comment. We do not have any changes in our financial disclosure.

Our answer:

Thank you for this comment. Laboratory protocol does not apply to our study, but study protocol does. Our study protocol has included in the Methods section.

Journal Requirements:

Our answer:

Thank you for this comment. We have ensured that our manuscript meets PLOS ONE's style requirements, including those for file naming.

2. We noted in your submission details that a portion of your manuscript may have been presented or published elsewhere. Please clarify whether this conference proceeding or publication was peer-reviewed and formally published. If this work was previously peer-reviewed and published, in the cover letter please provide the reason that this work does not constitute dual publication and should be included in the current manuscript.

Our answer:

This multicenter prospective observational study is the major update of our published previous study.[1] Moreover, we presented a portion of this manuscript as an abstract poster at the 14th World Stroke Congress organized by the World Stroke Organization, which took place in Singapore between 26 and 29 October 2022,[2] and only the poster abstract was peer-reviewed and formally published in the conference proceeding of the 14th World Stroke Congress.[3] However, we performed additional experiments or collected additional data that were not a part of the study from the published articles. We also presented new data in this submission that were not previously presented in the published articles. Therefore, this work does not constitute dual publication and should be included in the current manuscript.

[1] Luong CQ, Ngo HM, Hoang HB, Pham DT, Nguyen TA, Tran TA, Nguyen DN, Do SN, Nguyen MH, Vu HD, Vuong HTT, Mai TD, Nguyen AQ, Le KH, Dao PV, Tran TH, Vu LD, Nguyen LQ, Pham TQ, Dong HV, Nguyen HT, Nguyen CV, Nguyen AD. Clinical characteristics and factors relating to poor outcome in patients with aneurysmal subarachnoid hemorrhage in Vietnam: A multicenter prospective cohort study. PLoS One. 2021 Aug 13;16(8):e0256150. doi: 10.1371/journal.pone.0256150.

[2] Linh Quoc Nguyen, Tuan Anh Nguyen, Ton Duy Mai, et al. Validation of the accuracy of the modified World Federation of Neurosurgical Societies subarachnoid haemorrhage grading scale for predicting the outcomes of patients with aneurysmal subarachnoid haemorrhage. Paper abstract presented at: 14th World Stroke Congress; 26-29 October 2022, 2022; Singapore.

[3] 14th World Stroke Congress, Singapore, 26-29 October 2022. International Journal of Stroke 2022; 17: 3-288.

3. Please include a caption for figure 2.

Our answer:

Thank you for this comment. We have included a caption for Figure 2.

4. Please upload a copy of Figure 2, to which you refer in your text on page 20. If the figure is no longer to be included as part of the submission please remove all reference to it within the text.

Our answer:

Thank you for this comment. We have uploaded a copy of Figure 2.

Our answer:

Thank you for this comment. We have included captions for the Supporting Information files at the end of the manuscript and have updated any in-text citations to match accordingly.

Additional Editor Comments:

I would like to apologize again for the long wait. However, the diametrically different assessments of Reviewer 1 and Reviewer 2 made a third review necessary.

Our answer:

Thank the Editor and Reviewers so much for their support and for taking the time to leave their excellent reviews.

RESPONSE TO REVIEWERS

Reviewers' comments:

Reviewer #1:

This is a very interesting study concerning the validation of modified World Federation of Neurosurgical Societies (WFNS) subarachnoid haemorrhage grading scale in predicting the 90-day poor outcome defined as modified Rankin Scale scores of 4 to 6. The validation process was also correlated to WFNS and H&H scores. The result showed more accurate than those in WFNS and H&H scores. The manuscript is well written and should be published.

Our answer:

We thank you for the positive feedback. We also thank the Reviewer for taking the time to review our manuscript.

Reviewer #2: Thank you for submitting your manuscript.

Your multi center analysis of the 3 SAH grades (WFNS, mWFNS and Hunt and Hess) was thoroughly performed.

I however do have some comments.

Our answer:

We thank you for the positive feedback. We have carefully considered the Reviewer’s comments and suggestions and have revised our manuscript following each of these points.

1. It is stated that the WFNS is more complex to administer than the H&H scales because it requires the GCS and the motor function. First, I am not sure how it is more complicated than the H&H, since the GCS is a standard validated objective score that all neurosurgeon, and even emergency medicine doctors are all comfortable with. The H&H grade is very subjective. In addition, the WFNS score requires the GCS and the presence of a motor deficit only, which isn’t to complicated.

Our answer:

Thank you for pointing this out. We have rewritten the paragraph appropriately, as follows:

"Because the WFNS scale requires only an assessment of the Glasgow coma scale (GCS) and motor function, it may be easier to administer than the H&H scale." (Page 5, Lines 113-115)

2. How many patients were excluded because of the initial GCs was not scored? Looking at the chart of the referring hospital or even the ambulance chart would yield the GCS.

Our answer:

Thank you for pointing this out. We would also like to thank the Reviewer for this valuable comment and suggestion that could help us improve data collection and quality for our current study project. We will consider looking at the chart of the referring hospital or even the ambulance chart rather than that only upon admission to the participating hospitals to yield the Glasgow coma score (GCS).

In the present study, because there was a lack of electronic health record systems in our participating hospitals, all data were prospectively collected on the same unified case record forms (CRF) by representatives/investigators (i.e., fully trained clinicians and surgeons) and were entered into a database via the EpiData Entry software after the completion of data collection for later analysis. Data on patients for whom the initial GCS was unable to be scored or on patients who became lost at both 30- and 90-day follow-up visits would not be entered into a database by investigators. Although no patients became lost at 30 and 90 days of follow-up during the present study, we did not have data on how many patients for whom investigators could not score the initial GCS and excluded them from the present study. Thus, this factor has resulted in an implicit selection bias and an enrolment and inclusion incompletion of patients in the study database. As a result, our cohort is likely to be underestimated in the incidence rate of poor outcomes and deaths. We have provided further discussion concerning the limitations of the present study in the Discussion section (Pages 27-28, Lines 461-477; Page 32, Lines 564-575).

3. Why were patients lost at 90 day? If surgical or even treated by embolization, were follow ups not performed?

Our answer:

Thank you for this comment. In our study, all patients received follow-up visits or phone contacts up to 90 days post-enrolment. Because poor neurological function on day 90th after the ictus served as the primary outcome, data on patients who became lost at 90-day follow-up visits or phone contacts would not be entered into a database by investigators. However, no patients became lost at 90 days of follow-up during the present study.

4. The management of the aneurysmal SAH, let it be surgical or endovascular treatment, if very variable. The treatment option may play an important role in the outcome of the patient. Not having a standardized treatment option in these patients, and looking at the outcome is not adequate. This is a main caveat of the study.

Our answer:

Thank you for this comment. This study aimed to determine the relationship between the grades on the modified World Federation of Neurosurgical Societies (WFNS), WFNS, and Hunt and Hess (H&H) scales and the actual outcomes and to compare the accuracy of these scales in predicting the outcomes of patients with aneurysmal subarachnoid hemorrhage (SAH), regardless of the method of aneurysm repairs. We defined the primary and secondary outcomes as the 90- and 30-day poor neurological function, respectively. We determined the exposure variables as the modified WFNS, WFNS and H&H scales. All data elements required for calculating these scales at the time of admission were prospectively collected on a case record form (CRF) and entered into a database via the EpiData Entry software for later analysis. We also determined confounding factors as the variables of the baseline and clinical characteristics, the neuroimaging findings, the management, and the complications. However, we did not yet present the Methods section to reader in an intelligible fashion. Therefore, we have streamlined the items following the TRIPOD statement - the TRIPOD checklist - for reporting a study developing or validating a multivariable prediction model for diagnosis or prognosis.[1]

[1] Moons KG, Altman DG, Reitsma JB, Ioannidis JP, Macaskill P, Steyerberg EW, Vickers AJ, Ransohoff DF, Collins GS. Transparent Reporting of a multivariable prediction model for Individual Prognosis or Diagnosis (TRIPOD): explanation and elaboration. Ann Intern Med. 2015 Jan 6;162(1):W1-73. doi: 10.7326/M14-0698. PMID: 25560730.

5. It is stated that 415 patients presented to the study sites. How many were excluded because of the inadequate documentation of GCS? And how many and why where they lost to follow-up?

Our answer:

Thank you for pointing this out. In the present study, because there was a lack of electronic health record systems in our participating hospitals, all data were prospectively collected on the same unified case record forms (CRF) by representatives/investigators (i.e., fully trained clinicians and surgeons) and were entered into a database via the EpiData Entry software after the completion of data collection for later analysis. Data on patients for whom the initial GCS was unable to be scored or on patients who became lost at both 30- and 90-day follow-up visits would not be entered into a database by investigators. Although no patients became lost at 30 and 90 days of follow-up during the present study, we did not have data on how many patients for whom investigators could not score the initial GCS and excluded them from the present study. We have reworded the paragraph stating that “415 patients presented to the study sites” as follows:

"Data on 415 eligible patients with aneurysmal SAH were submitted to the study database (Fig. 1 and Table 1), in which there were few missing data." (Page 14, Lines 310-311)

6. The conclusion that the mWFSN score was strongest associated with an increased risk of poor outcome at 30/90 days of ictus is a strong statement. This needs to be revised accordingly to the limits of the study.

Our answer:

Thank you for pointing this out. We have reworded the Conclusion sections in the Abstract and the main text as follows:

“In this study, the modified WFNS, WFNS, and H&H scales all had good discriminatory abilities for the prognosis of patients with aneurysmal SAH. Because of the better effect size in predicting poor outcomes, the modified WFNS scale seems preferable to the WFNS and H&H scales.” (Page 4, Lines 80-83)

“This study investigated a selected cohort of patients with aneurysmal SAH, a high rate of poor outcomes and a high mortality rate presented to central hospitals in Vietnam. The modified WFNS, WFNS, and H&H scales all had good discriminatory abilities for the prognosis of patients with aneurysmal SAH. Because of the better effect size in predicting poor outcomes, the modified WFNS scale seems preferable to the WFNS and H&H scales.” (Page 32, Lines 578-582)

7. Although their data do support their conclusion, I do not believe that this has any clinical impact, nor does it help in the clinical decision making of the surgeon /radiologist.

Although interesting, the data inclusion criteria is very weak.

Our answer:

Thank the Reviewer so much for taking the time to leave an excellent review.

Reviewer #3:

The authors present the results of a prospective trial in patients with aneurysmal SAH, attempting to analyze the predictive value of SAH scales for poor outcome. This is an interesting study that carries high importance for the field. However, there are a number of aspects that need to be addressed before publication.

Our answer:

We thank you for the positive feedback. We have carefully considered the Reviewer’s comments and suggestions and have revised our manuscript following each of these points.

• How long was the median follow-up time?

Our answer:

Thank you for this comment. In the present study, all patients received a follow-up visit or phone contact till death in the hospital or within 30 or 90 days after symptom onset of aneurysmal subarachnoid hemorrhage (SAH) and had clinic visits or phone contacts on days 30th and 90th after ictus. We have clarified this issue in the Methods section (Page 7, Lines 154-158). Therefore, the median follow-up time was 30 (IQR: 15 - 20) days.

• Were there any changes of patient status after 90 days ?

Our answer:

In our study, all patients received a follow-up visit or phone contact till death in the hospital or within 30 or 90 days after symptom onset of aneurysmal subarachnoid hemorrhage (SAH) and had clinic visits or phone contacts on days 30th and 90th after ictus. Therefore, data on changes in patient status beyond over 90 days after onset was unavailable in our study.

• What was the outcome at discharge?

Our answer:

Thank you for this comment. In the present study, outcomes at hospital discharge were poor neurological function (poor outcome, defined as mRS of 4 to 6), as shown in S5 Table in S1 File.

• In the participants and treatment section, it is mentioned that “In the case of aphasia, patients were classified according to the clinically possible GCS scores derived from their eye and motor scores. How exactly is the “eye score” performed, and how was it implemented in the GS scale?

Our answer:

Thank you for this comment.

In the present study, all data elements required for calculating the subarachnoid hemorrhage (SAH) grading scales at the time of admission were prospectively collected on a case record form (CRF) and entered into a database via the EpiData Entry software for later analysis.

In the case of aphasia (3.2%; 10/415; kindly see S3 Table in S1 File for additional information), patients were classified according to the clinically possible Glasgow coma scale (GCS) scores derived from their eye and motor scores. For this purpose, the verbal scoring strategy in these patients was replaced by the median verbal score of patients with similar eye-motor scores but with lesions of the non-dominant hemisphere.[30],[31] This strategy involves the following steps: (a) First, A consecutive series of patients are assessed using the GCS. No verbal score is given to those with aphasia; (b) Second, All combinations of eye and motor scores for the patients without aphasia and with lesions of the non-dominant hemisphere are tabulated. The median verbal score is then determined for each combination; and (c) Finally, In patients with aphasia, the verbal score is imputed with the median verbal score of the patients without aphasia with lesions of the non-dominant hemisphere but with the same eye and motor scores.

We have cited the paragraph concerning the verbal scoring strategy in patients with missing verbal scores to two relevant references (Page 8, Line 169; Page 38, Lines 712-717), as follows:

References

[30] Juvela S. Risk factors for impaired outcome after spontaneous intracerebral hemorrhage. Arch Neurol. 1995 Dec;52(12):1193-200. doi: 10.1001/archneur.1995.00540360071018. PMID: 7492294.

[31] Prasad K, Menon GR. Comparison of the three strategies of verbal scoring of the Glasgow Coma Scale in patients with stroke. Cerebrovasc Dis. 1998 Mar-Apr;8(2):79-85. doi: 10.1159/000015822. PMID: 9548004.

• How were missing data handled?

Our answer:

Thank you for this comment. To minimize missing data, we performed the following steps: (a) First, we recorded data for each study patient in the same unified samples (case record form). A case record form was adopted across the study sites to collect the common variables; (b) Second, we submitted the data to the study database via EpiData Entry software, which was used for simple or programmed data entry and data documentation that could prevent data entry errors or mistakes, after the completion of data collection for later analysis; and (c) Finally, we checked the data for implausible outliers and missing fields and contacted hospital representatives for clarification.

As a result, there were few missing data in our study (S21 Table in S1 File). Therefore, we did not use any ways of handling missing values.

• Since the applied SAH grading systems are on an ordinal scale niveau, how meaningful is the approach of calculating ROC, which usually requires continuous data as the diagnostic input variable? For example, formulating a resulting cut-off value of 2.5 for poor outcomes in the modified WFNS scale is clinically challenging to implement. Please elaborate.

Our answer:

Thank you for this comment. In the present study, we converted from descriptive subarachnoid hemorrhage (SAH) grading scales to numerical SAH grading scales in ascending order (kindly see S1 Table in S1 File for additional information). Because of this, the cut-off value of each SAH grading scale was determined by receiver operator characteristic (ROC) curve analysis and defined as more or equal to the cut-off point with the maximum value of Youden’s index (i.e., sensitivity + specificity - 1). However, we have clarified this issue in the Statistical analyses section (Page 12, Lines 257-265) and throughout the manuscript, as follows:

"We converted from descriptive SAH grading scales to numerical SAH grading scales in ascending order (S1 Table in S1 file). Receiver operator characteristic (ROC) curves were plotted, and the areas under the ROC curve (AUROC) were calculated to determine the discriminatory ability of all SAH grading scales for the prognosis of the patients upon admission. The cut-off value of each SAH grading scale was determined by ROC curve analysis and defined as the cut-off point with the maximum value of Youden’s index (i.e., sensitivity + specificity - 1). Based on the cut-off value of each SAH grading scale, we assigned the patients to two severity groups: either the grade that was less than the cut-off value or another that was greater than or equal to the cut-off value." (Page 12, Lines 257-265)

• Comparing odd`s ratios between groups and models has been criticized, in fact, this approach has recently been rejected as an adequate method in this context. Several authors have argued that odds ratios will change if variables are added to the model, even if those additional variables are independent from the other variables. This concern particularly applies to the analysis of differences between clinical outcomes of the adjacent grades. Please comment on this critical aspect.

Our answer:

Thank you for this comment. The Reviewer is right that previous critiques have identified a caution problem with odds ratios,[1-5] and Edward and Bryan (2018) have recently raised this issue again.[6] Allison (1999) explained why odds ratios cannot be compared across samples.[1] Mood (2010) extended this work nicely to show that odds ratios cannot be interpreted as absolute effects, nor can they be compared across models or across groups within models.[2] Several authors have pointed out that odds ratios will change if variables are added to the model, even if those additional variables are independent of the other variables (Gail, Wieand, and Piantadosi 1984;[3] Yatchew and Griliches, 1985;[4] Allison 1999;[1] Mood 2010[2]). Mroz and Zayats (2008) also discussed the effect of omitted variables on the interpretation of odds ratios in logit models. Overall, a recent literature review (Edward and Bryan, 2018) has shown that there is no single odds ratio; instead, any estimated odds ratio is conditional on the data and the model specification.[6] Odds ratios should not be compared across different studies using different samples from different populations.[6] Nor should they be compared across models with different sets of explanatory variables.[6] Therefore, we have removed comparisons of odds ratios for poor outcomes among the intergrades of the subarachnoid hemorrhage (SAH) grading scales. Moreover, we have further discussed the comparison of odds ratios between grades for predicting poor outcomes in the Discussion section. (Pages 30-31, Lines 523-544)

References

[1] Allison PD. Comparing Logit and Probit Coefficients Across Groups. Sociological Methods & Research. 1999;28(2):186-208. doi: 10.1177/0049124199028002003.

[2] Mood C. Logistic Regression: Why We Cannot Do What We Think We Can Do, and What We Can Do About It. European Sociological Review. 2009;26(1):67-82. doi: 10.1093/esr/jcp006.

[3] Gail MH, Wieand S, Piantadosi S. Biased estimates of treatment effect in randomized experiments with nonlinear regressions and omitted covariates. Biometrika. 1984;71(3):431-44. doi: 10.1093/biomet/71.3.431.

[4] Yatchew A, Griliches Z. Specification Error in Probit Models. The Review of Economics and Statistics. 1985;67(1):134-9. doi: 10.2307/1928444.

[5] Mroz TA, Zayats YV. Arbitrarily Normalized Coefficients, Information Sets, and False Reports of “Biases” in Binary Outcome Models. The Review of Economics and Statistics. 2008;90(3):406-13. doi: 10.1162/rest.90.3.406.

[6] Norton EC, Dowd BE. Log Odds and the Interpretation of Logit Models. Health services research. 2018;53(2):859-78. doi: https://doi.org/10.1111/1475-6773.12712.

• Clinically, there are a number of variables that may influence the results of this analysis. In particular, was there an influence of aneurysm location and the frequency of vasospasm / delayed ischemia? How many patients received decompressive craniectomy?

Our answer:

Thank you for this comment. In the present study, we found that vertebral artery (VA) aneurysm and delayed cerebral ischemia (DCI) were significantly associated with poor outcomes on day 30th and 90th after ictus in the univariable logistic regression analyses. However, the multivariable logistic regression analysis showed only DCI was an independent predictor of poor outcomes on day 30th and 90th after ictus. We have added these variables to the multivariable logistic regression model (Tables 5 and 6, and S17 to S20 Tables in S1 File). In our study, surgical hematoma evacuation was defined as any surgical procedure evacuating epidural, subdural, intraventricular, or intraparenchymal hematoma, such as decompressive craniotomy, open craniotomy, or minimally invasive surgery. Of the total patients, 10.6% (44/415) received surgical hematoma evacuation. However, data on the reasons for surgical hematoma evacuation was unavailable in the present study. We have clarified this issue in the Methods section (Page 10, Lines 218-221) and have added these variables to S5 Table in S1 File.

• The strongest risk factor for poor outcome was the modified WFNS score of 3-5, which translates into an initial GCS score of 3-13. This variability makes the application of such a “risk - factor” somewhat challenging to implement in the clinical setting.

Our answer:

Thank you for this comment. In the present study, based on the cut-off value (≥2.5) of modified WFNS (Figs. 2 and 3), we assigned the patients to two severity groups: either the modified WFNS grade of I to II or another grade of III to V. In the multivariable logistic regression model, a modified WFNS grade of III to V was an independent predictor of the poor outcome (Tables 5 and 6). However, when we added the originally-suggested modified WFNS scale, with grade I taken as the reference, to the multivariable logistic regression model, with the same set of confounding variables, we found a gradual increase in adjusted odds ratio (AOR) of the modified WFNS scale, in ascending grades, for predicting the poor outcome (S11 and S14 in S1 File). These findings mean that a modified WFNS grade of III to V was an independent predictor of poor outcome, of which a higher modified WFNS grade was associated with a higher risk of poor outcome. Therefore, this variability does not make applying a “risk factor” challenging to implement in clinical settings (Kindly see Table 1 for additional information).

• To my best knowledge, the modified WFNS score appears to carry a better discriminatory value for good outcomes compared to the original WFNS score. Do the authors find similar results for the patients with a good outcome?

Our answer:

Thank you for this comment. To date, all SAH grading scales have been developed for predicting poor outcomes, in ascending grades, such as the Hunt and Hess (H&H) grading scale, World Federation of Neurosurgical Societies (WFNS) grading scale, or modified WFNS grading scale. It means that a higher grade on the SAH grading scales was associated with a higher risk of poor outcomes, and vice versa. A previously published study shows that the modified WFNS and the original WFNS scale both had good discriminatory ability concerning the prognosis of patients (either good or poor outcome) on day 90th after ictus, with the AUROC value of the modified WFNS scale that was significantly greater than those of the original WFNS scale.[1] In the present study, although the modified WFNS and the original WFNS scale both had good discriminatory ability concerning the prognosis of patients (either good or poor outcome) on day 90th after ictus (Fig. 3), there were no significant differences between the AUROC values of these scales (Table 2). This variation might be because of the differences concerning the outcome measures (i.e., the good outcome, defined as mRS score ≤1 in [1] vs defined as mRS score ≤3 in our study) between the two studies.

References

[1] Sano H, Satoh A, Murayama Y, Kato Y, Origasa H, Inamasu J, Nouri M, Cherian I, Saito N; members of the 38 registered institutions and WFNS Cerebrovascular Disease & Treatment Committee. Modified World Federation of Neurosurgical Societies subarachnoid hemorrhage grading system. World Neurosurg. 2015 May;83(5):801-7. doi: 10.1016/j.wneu.2014.12.032. Epub 2014 Dec 20. PMID: 25535068.

• Since the initial SAH grading is influenced by acute hydrocephalus, how many patients with acute hydrocephalus improved after implantation of an intraventricular drain? Did these patients show a different result regarding the SAH scales ad their prediction of poor outcomes?

Our answer:

Thank you for this comment. In our study, 32.8% (136/415) of patients with aneurysmal subarachnoid hemorrhage (SAH) had a complication of acute hydrocephalus, and 10.4% (43/414) received external ventricular drain (EVD). However, data on the reason for the insertion of an EVD was acute hydrocephalus which accounted for only 7.7% (32/414) of patients with aneurysmal SAH, and only 135 patients with aneurysmal SAH complicated by acute hydrocephalus was it recorded if an EVD was inserted or not, as shown in Table below. Of 135 patients with aneurysmal SAH complicated by acute hydrocephalus (Table below), 23.7% (32/135) of patients received an EVD; only 34.4% (11/32) of whom had the 30-day or 90-day good outcome, defined as a mRS score of 0 to 3. The table below also shows no significant difference in the admission severity and outcomes between patients who received an EVD and patients who did not. Therefore, the present study revealed that EVD did not improve poor outcomes in patients with aneurysmal SAH complicated by acute hydrocephalus, which might not impact the discriminatory ability of SAH grading scales for predicting poor outcomes.

Table: The admission severity and the outcome among patients with aneurysmal subarachnoid hemorrhage complicated by acute hydrocephalus

(See 'Response to Reviewers' file attached)

Thank the Editors and Reviewers so much for taking the time to leave their excellent reviews!

Sincerely yours,

Chinh Quoc Luong, MD., PhD.

Center for Emergency Medicine,

Bach Mai Hospital,

No. 78, Giai Phong, Phuong Mai ward, Dong Da district, Hanoi 100000, Vietnam

Email: luongquocchinh@gmail.com

---

## [Decision Letter · Decision Letter 1]

9 Jun 2023

PONE-D-22-32313R1Validation of the accuracy of the modified World Federation of Neurosurgical Societies Subarachnoid Hemorrhage Grading Scale for predicting the outcomes of patients with aneurysmal subarachnoid hemorrhagePLOS ONE

Dear Dr. Luong,

Thank you for submitting your manuscript to PLOS ONE. After careful consideration, we feel that it has merit but does not fully meet PLOS ONE’s publication criteria as it currently stands. Therefore, we invite you to submit a revised version of the manuscript that addresses the points raised during the review process.

We look forward to receiving your revised manuscript.

Kind regards,

Martin Kieninger

Academic Editor

PLOS ONE

Journal Requirements:

Reviewers' comments:

Reviewer's Responses to Questions

**Comments to the Author**

1. If the authors have adequately addressed your comments raised in a previous round of review and you feel that this manuscript is now acceptable for publication, you may indicate that here to bypass the “Comments to the Author” section, enter your conflict of interest statement in the “Confidential to Editor” section, and submit your "Accept" recommendation.

Reviewer #3: All comments have been addressed

Reviewer #4: All comments have been addressed

2. Is the manuscript technically sound, and do the data support the conclusions?

Reviewer #3: Yes

Reviewer #4: Yes

3. Has the statistical analysis been performed appropriately and rigorously? 

Reviewer #3: Yes

Reviewer #4: Yes

4. Have the authors made all data underlying the findings in their manuscript fully available?

Reviewer #3: Yes

Reviewer #4: Yes

5. Is the manuscript presented in an intelligible fashion and written in standard English?

Reviewer #3: Yes

Reviewer #4: Yes

6. Review Comments to the Author

Reviewer #3: The authors have in fact carefully addressed all aspects I have raised in my review. I congratulate the authors for this important study

Reviewer #4: Dear Dr. Luong. You and the manuscript writing team have made a great and highly precious work. I am really impressed by the study methodology as compared to the original WFNS, HH and MWFNS comparative reviews. The inclusion of all radiologic and epidemiological and clinical data, the transparent way of sample calculation and cut-off estimation as well as the logistic regression analysis were highly precise and well-performed. However, I would like the inclusion of 2 items to enhance the value of this great work:

1- inclusion of PAASH scale in the calculation for ROC analysis curve and the cut-off prognostication table. the 5-category Prognosis on Admission of Aneurysmal Subarachnoid Hemorrhage (PAASH) grading scale has been shown by S.M. Dorhout Mees et al [DOI: 10.1161/STROKEAHA.107.498345] to show a more gradual increase of OR in ascending categories as compared to WFNS scale and as comparable to modified WFNS. In your work, your stated that " the modified WFNS scale did not show more gradual increases in OR for the 30- and 90-day poor outcome, in ascending grades, compared to the WFNS and H&H scales " , so inclusion of PAASH gradual OR prognostication to this context will render your manuscript more comprehensive and recognizable. Additionally, the mRS 4-5 were defined by you as a poor outcome, though some papers include mRS 3 as well. An explanation for the situation of mRS 3 patient categories will be highly valuable. These notes don't demerit your valuable work. Rather, they ensure more worldwide recognition of it. Thank you.

7. PLOS authors have the option to publish the peer review history of their article (what does this mean?). If published, this will include your full peer review and any attached files.

Reviewer #3: No

Reviewer #4: **Yes: **Mohamed Mostafa

---

## [Author Response · Author response to Decision Letter 1]

24 Jun 2023

Professor Martin Kieninger

Academic Editor

PLOS ONE

June 24, 2023

Dear Prof. Martin Kieninger,

On behalf of all authors, I am resubmitting herewith our revised manuscript entitled “Validation of the accuracy of the modified World Federation of Neurosurgical Societies Subarachnoid Hemorrhage Grading Scale for predicting the outcomes of patients with aneurysmal subarachnoid hemorrhage” (PONE-D-22-32313R2).

We sincerely appreciate the kind comments and points raised by the Editors and by the Reviewers. We have carefully considered all comments and suggestions and revised our manuscript following each of these points. These comments have enabled us to substantially improve our manuscript. We hope that Editor will find our revised manuscript suitable for publication in PLOS ONE.

We confirm that this work is original and has not been published elsewhere nor is it currently under consideration for publication elsewhere. All authors have read, approved the manuscript, and agreed to authorship and order of authorship for this manuscript, and all authors have the appropriate permissions and rights to the reported data.

We have provided our point-by-point responses to the comments of the Editors and the Reviewers attached.

We thank you for your kind consideration of this submission.

Sincerely yours,

Chinh Quoc Luong, MD., PhD.

Center for Emergency Medicine,

Bach Mai Hospital,

No. 78, Giai Phong, Phuong Mai ward, Dong Da district, Hanoi 100000, Vietnam

Email: luongquocchinh@gmail.com

We thank the Editors and the Reviewers for the valuable comments and suggestions that greatly helped us to improve the contents of this paper. In what follows, we will use the boldface to indicate comments from the Editors and the Reviewers, the standard font face for our responses and we highlighted in yellow the modifications that we did to the manuscript.

RESPONSE TO EDITORS

Thank you for submitting your manuscript to PLOS ONE. After careful consideration, we feel that it has merit but does not fully meet PLOS ONE’s publication criteria as it currently stands. Therefore, we invite you to submit a revised version of the manuscript that addresses the points raised during the review process.

Our answer:

We thank you for the positive feedback. We have carefully considered the Reviewers' comments and suggestions and have revised our manuscript following each of these points.

Thank you for this comment. We have submitted the revised manuscript on time.

Our answer:

Thank you for this comment. We have included a rebuttal letter, a marked-up copy of the manuscript, and an unmarked version of the revised manuscript when submitting our revised manuscript.

Our answer:

Thank you for this comment. We do not have any changes in our financial disclosure.

Our answer:

Thank you for this comment. Laboratory protocol does not apply to our study, but study protocol does. Our study protocol has included in the Methods section.

We look forward to receiving your revised manuscript.

Kind regards,

Martin Kieninger

Academic Editor

PLOS ONE

Our answer:

Thank the Editor and Reviewers so much for their support and for taking the time to leave their excellent reviews.

Journal Requirements:

Our answer:

Thank you for this comment. We have reviewed the reference list, which is complete and correct. There are not any retracted articles on the list.

RESPONSE TO REVIEWERS

Reviewers' comments:

Reviewer #3: The authors have in fact carefully addressed all aspects I have raised in my review. I congratulate the authors for this important study

Our answer:

Thank the Reviewer so much for taking the time to leave his/her excellent review.

Reviewer #4: Dear Dr. Luong. You and the manuscript writing team have made a great and highly precious work. I am really impressed by the study methodology as compared to the original WFNS, HH and MWFNS comparative reviews. The inclusion of all radiologic and epidemiological and clinical data, the transparent way of sample calculation and cut-off estimation as well as the logistic regression analysis were highly precise and well-performed. However, I would like the inclusion of 2 items to enhance the value of this great work:

Our answer:

Thank the Reviewer so much for taking the time to leave his/her excellent review. We have carefully considered the Reviewer’s comments and suggestions and have revised our manuscript following each of these points.

1- inclusion of PAASH scale in the calculation for ROC analysis curve and the cut-off prognostication table. the 5-category Prognosis on Admission of Aneurysmal Subarachnoid Hemorrhage (PAASH) grading scale has been shown by S.M. Dorhout Mees et al [DOI: 10.1161/STROKEAHA.107.498345] to show a more gradual increase of OR in ascending categories as compared to WFNS scale and as comparable to modified WFNS.

Our answer:

Thank you for this valuable comment and suggestion. This study is part of an ongoing research project that aims to develop and validate the accuracy of several subarachnoid hemorrhage (SAH) grading scales based on admission Glasgow Coma Score (GCS) for predicting the outcomes of patients with aneurysmal SAH. With the latest updated dataset, we have performed further analysis and found that the 5-category Prognosis on Admission of Aneurysmal Subarachnoid Hemorrhage (PAASH) grading scale showed a more gradual increase of odds ratio in ascending categories compared to the Hunt and Hess (H&H), World Federation of Neurological Surgeons (WFNS), and modified WFNS grading scales. We will report the result in another paper.

In your work, your stated that " the modified WFNS scale did not show more gradual increases in OR for the 30- and 90-day poor outcome, in ascending grades, compared to the WFNS and H&H scales ", so inclusion of PAASH gradual OR prognostication to this context will render your manuscript more comprehensive and recognizable. Additionally, the mRS 4-5 were defined by you as a poor outcome, though some papers include mRS 3 as well. An explanation for the situation of mRS 3 patient categories will be highly valuable. These notes don't demerit your valuable work. Rather, they ensure more worldwide recognition of it. Thank you.

Our answer:

Thank you for your valuable comment and suggestion. This study is part of an ongoing research project that aims to develop and validate the accuracy of several subarachnoid hemorrhage (SAH) grading scales based on admission Glasgow Coma Score (GCS) for predicting the outcomes of patients with aneurysmal SAH. In this study, we defined good outcomes as a modified Rankin Scale (mRS) score of 0 to 3, which is also used in previously published studies.[1],[2] However, with the latest updated dataset, we will define excellent outcomes as an mRS score of 0 to 2 and report the result in another paper.

[1] Hoh BL, Ko NU, Amin-Hanjani S, Hsiang-Yi Chou S, Cruz-Flores S, Dangayach NS, Derdeyn CP, Du R, Hänggi D, Hetts SW, Ifejika NL, Johnson R, Keigher KM, Leslie-Mazwi TM, Lucke-Wold B, Rabinstein AA, Robicsek SA, Stapleton CJ, Suarez JI, Tjoumakaris SI, Welch BG. 2023 Guideline for the Management of Patients With Aneurysmal Subarachnoid Hemorrhage: A Guideline From the American Heart Association/American Stroke Association. Stroke. 2023 May 22. doi: 10.1161/STR.0000000000000436.

[2] Post R, Germans MR, Tjerkstra MA, Vergouwen MDI, Jellema K, Koot RW, Kruyt ND, Willems PWA, Wolfs JFC, de Beer FC, Kieft H, Nanda D, van der Pol B, Roks G, de Beer F, Halkes PHA, Reichman LJA, Brouwers PJAM, van den Berg-Vos RM, Kwa VIH, van der Ree TC, Bronner I, van de Vlekkert J, Bienfait HP, Boogaarts HD, Klijn CJM, van den Berg R, Coert BA, Horn J, Majoie CBLM, Rinkel GJE, Roos YBWEM, Vandertop WP, Verbaan D; ULTRA Investigators. Ultra-early tranexamic acid after subarachnoid haemorrhage (ULTRA): a randomised controlled trial. Lancet. 2021 Jan 9;397(10269):112-118. doi: 10.1016/S0140-6736(20)32518-6.

Thank the Editors and Reviewers so much for taking the time to leave their excellent reviews!

Sincerely yours,

Chinh Quoc Luong, MD., PhD.

Center for Emergency Medicine,

Bach Mai Hospital,

No. 78, Giai Phong, Phuong Mai ward, Dong Da district, Hanoi 100000, Vietnam

Email: luongquocchinh@gmail.com

---

## [Decision Letter · Decision Letter 2]

17 Jul 2023

Validation of the accuracy of the modified World Federation of Neurosurgical Societies Subarachnoid Hemorrhage Grading Scale for predicting the outcomes of patients with aneurysmal subarachnoid hemorrhage

PONE-D-22-32313R2

Dear Dr. Dao,

We’re pleased to inform you that your manuscript has been judged scientifically suitable for publication and will be formally accepted for publication once it meets all outstanding technical requirements.

Kind regards,

Martin Kieninger

Academic Editor

PLOS ONE

Additional Editor Comments (optional):

Reviewers' comments:

Reviewer's Responses to Questions

**Comments to the Author**

1. If the authors have adequately addressed your comments raised in a previous round of review and you feel that this manuscript is now acceptable for publication, you may indicate that here to bypass the “Comments to the Author” section, enter your conflict of interest statement in the “Confidential to Editor” section, and submit your "Accept" recommendation.

Reviewer #4: All comments have been addressed

2. Is the manuscript technically sound, and do the data support the conclusions?

Reviewer #4: Yes

3. Has the statistical analysis been performed appropriately and rigorously? 

Reviewer #4: Yes

4. Have the authors made all data underlying the findings in their manuscript fully available?

Reviewer #4: Yes

5. Is the manuscript presented in an intelligible fashion and written in standard English?

Reviewer #4: Yes

6. Review Comments to the Author

Reviewer #4: I thank editors for their valuable consideration of my review points and would lije to express my apologies for late response due to urgent circumstances. Thank you for sharing me this spectacular chance.

7. PLOS authors have the option to publish the peer review history of their article (what does this mean?). If published, this will include your full peer review and any attached files.

Reviewer #4: **Yes: **Mohamed Mostafa

---

## [Editor Report · Acceptance letter]

11 Aug 2023

PONE-D-22-32313R2 

Validation of the accuracy of the modified World Federation of Neurosurgical Societies Subarachnoid Hemorrhage Grading Scale for predicting the outcomes of patients with aneurysmal subarachnoid hemorrhage 

Dear Dr. Dao:

I'm pleased to inform you that your manuscript has been deemed suitable for publication in PLOS ONE. Congratulations! Your manuscript is now with our production department. 

Kind regards, 

on behalf of

Dr. Martin Kieninger 

Academic Editor

PLOS ONE